# Molecular mechanism of antibody neutralization of coxsackievirus A16

Chao Zhang ®[1,2,4], Caixuan Liu[3,4], Jinping Shi[1,4], Yalei Wang[1], Cong Xu[3], Xiaohua Ye[1], Qingwei Liu[1], Xue Li[1], Weihua Qiao[1], Yannan Yin[1], Yao Cong ®[3] ✉ & Zhong Huang ®[1,2] ✉

Coxsackievirus A16 (CVA16) causes hand, foot and mouth disease in infants and young children. However, no vaccine or anti-viral agent is currently available for CVA16. Here, the functions and working mechanisms of two CVA16-specific neutralizing monoclonal antibodies (MAbs), 9B5 and 8C4, are comprehensively investigated. Both 9B5 and 8C4 display potent neutralization in vitro and prophylactic and therapeutic efficacy in a mouse model of CVA16 infection. Mechanistically, 9B5 exerts neutralization primarily through inhibiting CVA16 attachment to cell surface via blockade of CVA16 binding to its attachment receptor, heparan sulfate, whereas 8C4 functions mainly at the post-attachment stage of CVA16 entry by interfering with the interaction between CVA16 and its uncoating receptor SCARB2. Cryo-EM studies show that 9B5 and 8C4 target distinct epitopes located at the 5-fold and 3-fold protrusions of CVA16 capsids, respectively, and exhibit differential binding preference to three forms of naturally occurring CVA16 particles. Moreover, 9B5 and 8C4 are compatible in formulating an antibody cocktail which displays the ability to prevent virus escape seen with individual MAbs. Together, our work elucidates the functional and structural basis of CVA16 antibody-mediated neutralization and protection, providing important information for design and development of effective CVA16 vaccines and antibody therapies.

Coxsackievirus A16 (CVA16), a member of the *Enterovirus genus* within the *Picornaviridae* family[1,2], is one of the major causative agents of hand, foot and mouth disease (HFMD) prevalent in infants and young children[3–6]. CVA16 infection may result in mild and self-limiting symptoms[7] as well as severe clinical outcomes such as encephalitis, myocarditis, pneumonitis, and even death[8–10]. In addition, CVA16 often co-circulates with other HFMD-causing agents such as human enterovirus A71 (EV71), coxsackievirus A6 (CVA6), and coxsackievirus A10 (CVA10), leading to co-infection and viral genetic recombination[11–13], which make it more challenging to prevent and control HFMD as a whole.

Like other enteroviruses (EVs), CVA16 is a nonenveloped virus of ~30 nm in diameter with a single-stranded, positive-sense RNA genome of ~7.4 kb in length packaged in a protein shell termed capsid[14]. The viral genome encodes a large polyprotein precursor, which is subsequently processed into structural protein P1 and nonstructural proteins P2 and P3[14]. P1 can be further cleaved by a viral protease to yield capsid subunit proteins VP0, VP1, and VP3, among which VP0 may undergo autocleavage to produce VP2 and VP4[14]. CVA16 prepared from infected cell cultures is present mainly in two particle forms, including the mature virion (also termed full particle) containing infectious viral RNA genome and the empty particle (also termed

---

[1]CAS Key Laboratory of Molecular Virology & Immunology, Institut Pasteur of Shanghai, Chinese Academy of Sciences, University of Chinese Academy of Sciences, Shanghai, China. [2]Shanghai Institute of Infectious Disease and Biosecurity, Fudan University, Shanghai, China. [3]State Key Laboratory of Molecular Biology, Shanghai Institute of Biochemistry and Cell Biology, Center for Excellence in Molecular Cell Science, Chinese Academy of Sciences, Shanghai, China. [4]These authors contributed equally: Chao Zhang, Caixuan Liu, Jinping Shi. ✉e-mail: cong@sibcb.ac.cn; huangzhong@ips.ac.cn

procapsid) lacking viral RNA genome[15–17]. The capsids of CVA16 full particle and empty particle are structurally similar, both made of 60 copies of protomers arranged in symmetry, however, each protomer in the full particle consists of four subunits, including VP1, VP2, VP3, and VP4, whereas the one in the empty particle comprises VP1, VP3, and uncleaved VP0[15–17]. For most EVs, their mature virions are in the native/compact state, whereas their empty particles appear relatively expanded[17–20]. However, empty particles in compact state have also been reported for CVA16 and hepatitis A virus[16,21].

CVA16 interacts with its host receptors to gain entry into susceptible cells. Specifically, CVA16 utilizes cell surface heparan sulfate (HS) glycosaminoglycans as its attachment receptor[22], and human scavenger receptor B2 (SCARB2, also known as lysosomal integral membrane protein 2, LIMP-2) as its uncoating receptor[23]. Upon receptor binding or biochemical treatment, CVA16 mature virions may transform to an uncoating intermediate state, termed the "135S-like particle" or "A-particle", with typical structural features such as an expanded capsid, loss of pocket factor, and an enlarged two-fold opening[24,25].

Thus far, neither preventive vaccines nor therapeutic drugs for CVA16 have been licensed for human use[4]. Neutralizing antibodies play a critical role in anti-viral protective immunity[26]. It is hence important to identify CVA16-specific neutralizing antibodies and determine their functionality, binding epitopes, and working mechanisms, which in turn may aid the development of anti-CVA16 vaccines and therapeutics. Previous studies have demonstrated that passive transfer of neutralizing antisera induced by CVA16 vaccine candidates, such as inactivated whole virus vaccines or recombinant virus like particles, could effectively protect mice against lethal CVA16 challenge[27–29]. A recent report showed that two CVA16-specific neutralizing monoclonal antibodies (MAb), namely 14B10 and NA9D7, displayed therapeutic efficacy in a mouse model of CVA16 infection[17]. A number of CVA16 neutralizing antibody epitopes have been identified through peptide ELISA screens or structural studies[17,30]. However, it remains elusive how antibodies targeting these epitopes/sites exert neutralization and protection. Especially, whether and how neutralizing antibodies affect CVA16 virus interacting with its receptors has not been explored.

In this work, we comprehensively characterize two CVA16-specific mouse neutralizing MAbs, designated 9B5 and 8C4, respectively. Both 9B5 and 8C4 antibodies display potent prophylactic and therapeutic efficacy in a mouse model of CVA16 infection. Mechanistic studies reveal that 9B5 and 8C4 target non-overlapping epitopes on the CVA16 viral capsid and adopt distinct mechanisms to exert neutralization. These two MAbs are compatible in formulating an antibody cocktail with increased neutralizing potency and, more importantly, able to prevent virus escape seen with individual antibodies. Our findings may have important implications for CVA16 vaccine and antibody development.

## Results

### Identification of neutralizing MAbs against CVA16

To prepare CVA16-specific neutralizing MAbs, hybridomas were generated from mice immunized with inactivated CVA16 (strain SZ05) and were then screened for neutralization against the parental virus CVA16/SZ05. Finally, three stable hybridoma mono-clones (8C4, 9B5, and 9G1) showing neutralizing activity were obtained (Fig. 1a). Based on cytopathic effect (CPE) observation, the neutralization concentrations (the lowest antibody concentration that fully protects cells from CPE) for MAbs 8C4, 9B5, and 9G1 were determined to be 313, 1.2, and 20 ng/ml, respectively (Fig. 1a). When analyzed using the cell viability assay, MAbs 8C4, 9B5, and 9G1 exhibited neutralization activities against CVA16/SZ05 in an antibody dose-dependent fashion (Fig. 1b), with half-maximal inhibitory concentrations (IC50) of 74, 0.4, and 4.9 ng/ml, respectively (Fig. 1a); in contrast, the IgG control antibody, 2H2[31], did not exhibit any neutralization in any of the tested

concentrations (Fig. 1b). Isotyping assay revealed that 8C4 and 9B5 belonged to IgG2b, while 9G1 was of IgG2a isotype (Fig. 1a). Antibody sequencing showed that variable regions of the three clones were derived from different VH and VL (heavy and light chain variable regions) gene families (Supplementary Fig. 1), indicating that 8C4, 9B5, and 9G1 were three distinct clones.

The three anti-CVA16 MAbs were further tested for their ability to cross-neutralize two heterologous CVA16 strains (CVA16/GX08 and CVA16/MAV [mouse-adapted virus], Supplementary Table 1) and one EV71 strain EV71/G082. MAbs 8C4, 9B5, and 9G1 were still able to potently neutralize CVA16/GX08 with neutralization concentrations (100% protection) of 1250, 4.9, and 78 ng/ml, respectively (Fig. 1a). IC50 of 8C4, 9B5, and 9G1 against CVA16/GX08 were determined to be 55, 1.2, and 7.9 ng/ml, respectively (Fig. 1a and Supplementary Fig. 2a). Similarly, neutralization concentrations of 8C4, 9B5, and 9G1 against CVA16/MAV were determined to be 1250, 4.9, and 78 ng/ml, respectively. IC50 of 8C4, 9B5, and 9G1 against CVA16/MAV were determined to be 52, 2.4, and 17 ng/ml, respectively (Fig. 1a and Supplementary Fig. 2b). All the three anti-CVA16 MAbs failed to neutralize EV71/G082 even at the highest concentration tested (100 μg/ml), whereas, as expected, the previously identified EV71-specific MAb D5[32] was highly effective in neutralizing EV71/G082 (Fig. 1a, c).

### Binding properties of the anti-CVA16 MAbs

We analyzed the binding affinity of individual anti-CVA16 MAbs to purified CVA16 viral particles by performing bio-layer interferometry (BLI) assay. Briefly, biotinylated CVA16/SZ05 viral particles were immobilized onto the sensors and then allowed to interact with different concentrations of individual MAbs. As shown in Fig. 1d, all of the three MAbs exhibited high binding affinity to CVA16/SZ05 viral particles with equilibrium dissociation constants (KD) being 7.34, 0.35, and 0.46 nM for 8C4, 9B5, and 9G1, respectively.

Next, we assessed whether the three MAbs competed with each other in binding CVA16 particles by performing a competition BLI assay. In this assay, immobilized CVA16/SZ05 viral particles were pre-incubated with kinetics buffer (reference) or the first antibody and then allowed to bind the second MAb in the presence of the first MAb (which can prevent the reduction of 1st antibody signal in the competition phase). Binding signals of the second MAb were calculated and shown in Fig. 1e–g. Compared with buffer alone, pre-incubation with the control IgG antibody 2H2 did not affect the binding of 8C4, 9B5, or 9G1 to CVA16 virion. Pre-incubation of either 9B5 or 9G1 with CVA16 viral particles resulted in a slight decrease in 8C4 binding signal (Fig. 1e), indicating minimal or no competition between 8C4 and 9B5/9G1. In line with this observation, it was found that pre-binding of MAb 8C4 led to very slight decrease in 9B5- and 9G1-binding signals (Fig. 1f, g). These data show that 8C4 epitope does not overlap with the 9B5- and 9G1-binding sites. In contrast, pre-binding of MAb 9G1 significantly blocked subsequent 9B5 binding, as indicated by a >50% decrease in BLI signal (Fig. 1f). Notably, pre-incubation of MAb 9B5 almost completely blocked subsequent 9G1 binding (Fig. 1g). Together, these data show that the binding sites of 9B5 and 9G1 on the CVA16 particle surface are at least partially overlapping. Based on the binding competition results, the three MAbs can be categorized into two non-competing antibody groups: the group 1 consists of 9B5 and 9G1, while 8C4 alone constitutes the group 2. MAb 9B5, which is a much more potent neutralizer than 9G1 in the group 1, and 8C4 in the group 2, were selected as the representatives of the two antibody groups for subsequent in-depth analyses.

### Prophylactic and therapeutic efficacies of 8C4 and 9B5

The protective effects of MAbs 8C4 and 9B5 were assessed in a previously reported mouse model of CVA16 infection, which was based on the mouse-adapted strain CVA16/MAV[28]. Besides individual MAbs, a

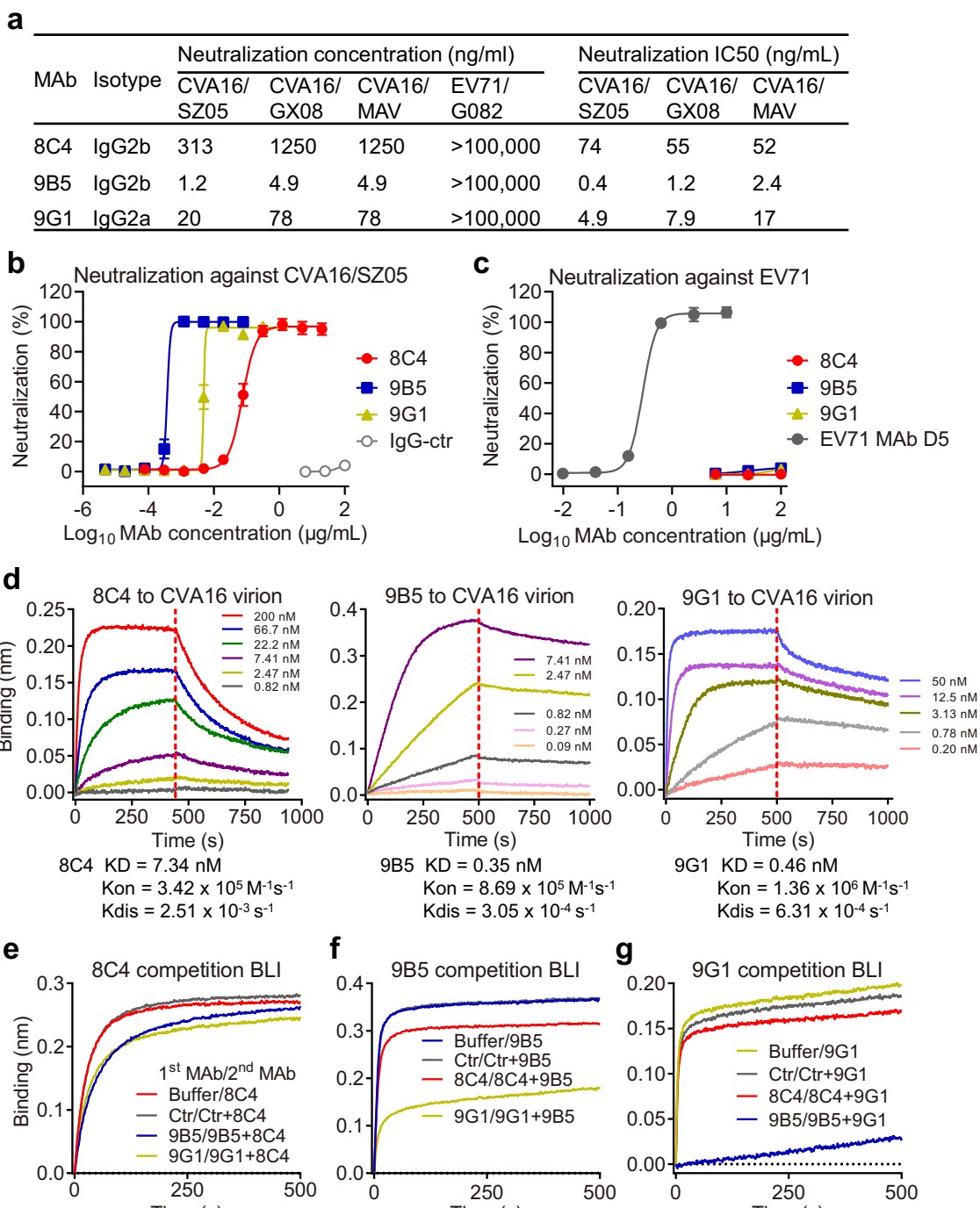

**Fig. 1 | Neutralization, binding affinity and mutual competition of anti-CVA16 MAbs. a** Isotypes and neutralization of anti-CVA16 MAbs (8C4, 9B5, and 9G1). Neutralization concentration was defined as the lowest antibody concentration that fully prevented cytopathic effect. Neutralization IC50 of each MAb against CVA16 was determined by the cell viability assay. **b** Neutralization of the MAbs against CVA16 strain SZ05 was measured by the cell viability assay. Anti-SARS-CoV-2 MAb 2H2 served as negative control (IgG-ctr). **c** Neutralization of the MAbs against EV71. Anti-EV71 MAb D5 served as positive control in this assay. In **b**, **c**, data are mean ± SEM of five replicate wells in 96-well cell culture plates. **d** Binding kinetics of the MAbs to immobilized CVA16/SZ05 viral particles were measured by bio-layer interferometry (BLI). Association and dissociation steps are divided by dotted red line. MAb concentrations used were shown. **e**–**g** BLI-based antibody competition assay. Immobilized CVA16/SZ05 viral particles were first incubated with buffer (reference) or the indicated MAb (first antibody) and then incubated with the second MAb 8C4 (**e**), 9B5 (**f**), or 9G1 (**g**) in the presence of the first MAb. An irrelevant MAb 2H2 served as the control antibody (gray curve) in the assay. The graphs show binding signals of the second MAb 8C4 (**e**), 9B5 (**f**), and 9G1 (**g**).

combination of 8C4 and 9B5 at a ratio of 1:1 was also included in the in vivo protection studies, considering that the two antibodies recognize non-overlapping epitopes on viral capsid and thus can be used together. For the prophylaxis study, groups of naïve 1-day-old ICR mice were injected with PBS, 10 mg/kg of anti-CVA16 MAbs 8C4 or 9B5, the combination of 8C4 and 9B5 (10 mg/kg of each antibody), or 10 mg/kg of the IgG control antibody 2H2[31]. Twenty-four hour later, the mice were challenged with CVA16/MAV. Survival and clinical scores were recorded daily and were depicted in Fig. 2a. Mice receiving PBS or IgG control antibody started to exhibit characteristic clinical

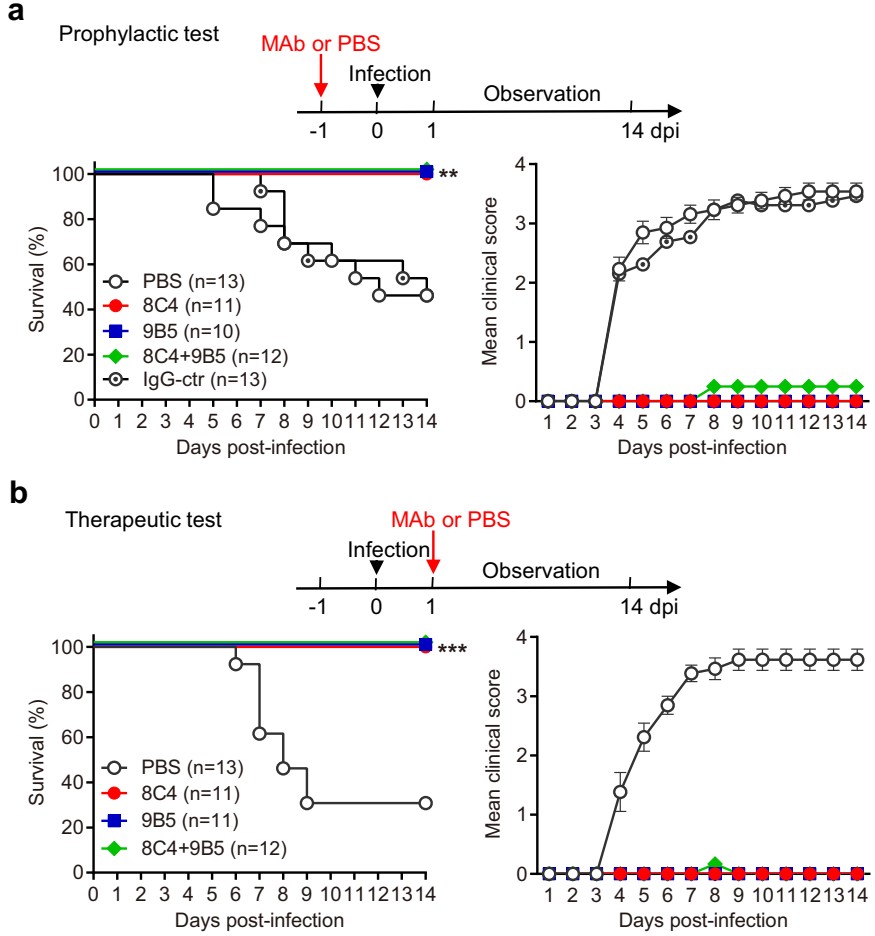

**Fig. 2 | The in vivo protective efficacy of the anti-CVA16 MAbs.** In vivo prophylactic efficacy (**a**) and therapeutic efficacy (**b**) of MAbs 8C4, 9B5 and the 8C4 + 9B5 cocktail against CVA16 infection in mice. Upper panel: study outline. Lower panel: survival and clinical score. Clinical scores were graded as follows: 0, healthy; 1, reduced mobility; 2, limb weakness; 3, limb paralysis; 4, death. The number of mice in each group were indicated in the bracket. Survival rates of MAb-treated mice were compared with the mice in the PBS-treated group. Statistical significance was determined by Log-rank (Mantel–Cox) test. \*\**p* < 0.01; \*\*\**p* < 0.001. For **a**, *p* value between the PBS group and the 8C4 group is 0.0048; *p* value between the PBS group and the 9B5 group is 0.0071; *p* value between the PBS group and the 8C4 + 9B5 group is 0.0032. For **b**, *p* value between the PBS group and the 8C4 or 9B5 group is 0.0006; *p* value between the PBS group and the 8C4 + 9B5 group is 0.0004. All error bars represent SEM. Note that the results from two independent experiments are shown in this figure and Supplementary Fig. 3, respectively.

symptoms, including reduced mobility, limb weakness and paralysis, at 4 days post-infection (dpi), and 54% of the mice in the two groups eventually died. In contrast, all of the mice treated with 8C4, 9B5, or the 8C4 + 9B5 combination were completely protected from death, and all of these mice, except one, showed no clinical signs of illness. These data demonstrate the potent in vivo prophylactic efficacy of 8C4, 9B5, and the 8C4 + 9B5 combination.

To evaluate the therapeutic efficacy of the anti-CVA16 MAbs, groups of 2-day-old ICR mice were infected with CVA16/MAV and 1 day later treated with PBS, 10 mg/kg of MAb 8C4, 10 mg/kg of MAb 9B5, or the 8C4 + 9B5 combination (10 mg/kg of each MAb). Survival and clinical scores were monitored daily (Fig. 2b). Mice given PBS developed obvious clinical signs of illness at 4 dpi and showed significant mortality rate (70%) by 14 dpi. In contrast, mice administered 8C4, 9B5, or the 8C4 + 9B5 combination showed 100% survival rates at 14 dpi, and none of the mice displayed any signs of infection during the course of the study. These results clearly demonstrate that 8C4, 9B5, and the 8C4 + 9B5 combination have extraordinary therapeutic efficacies.

To confirm the in vivo efficacy of the anti-CVA16 MAbs, both prophylactic and therapeutic tests were repeated and similar results were obtained (Supplementary Fig. 3). All of the mice administered anti-CVA16 MAbs, except one 8C4-treated mouse in the therapeutic test, were completely protected. In contrast, the mice in the PBS or the control IgG groups displayed significant mortality ranging from 46 to 67% (Supplementary Fig. 3c). Collectively, the above data demonstrate that 8C4 and 9B5 possess robust prophylactic and therapeutic efficacies in vivo.

## Neutralization mechanisms for MAbs 8C4 and 9B5

To understand the mechanisms of CVA16 neutralization and protection mediated by MAbs 8C4 and 9B5, we firstly performed time-of-addition assay to determine at which step the MAbs exert anti-viral effects. Briefly, cultured cells were either inoculated with CVA16 that had been pre-mixed with individual MAbs (pre-attachment) or the cells were treated with the MAbs at different time points after virus attachment at 4 °C (post-attachment). Then, the cells were analyzed for relative viral RNA levels at 6 h post-infection (hpi). As shown in Fig. 3a, the addition of antibody 8C4 before virus adsorption (Pre) or at 0 or 0.5 h after virus-bound cells were shifted to 37 °C (Post-0 h, Post-0.5 h) resulted in significantly reduced viral RNA levels as compared to the virus-only group; the relative viral RNA levels were determined to be 23.2%, 29.8%, and 49.6% for the 8C4 Pre, Post-0 h, and Post-0.5 h groups, respectively. It was noted that the relative viral RNA levels in the 8C4 Post-0.5 h group were significantly higher than those of the Pre or the Post-0 h samples, indicating reduced neutralization effect by 8C4 administered at 0.5 h post-attachment. These results suggest that 0.5 h post-attachment is a

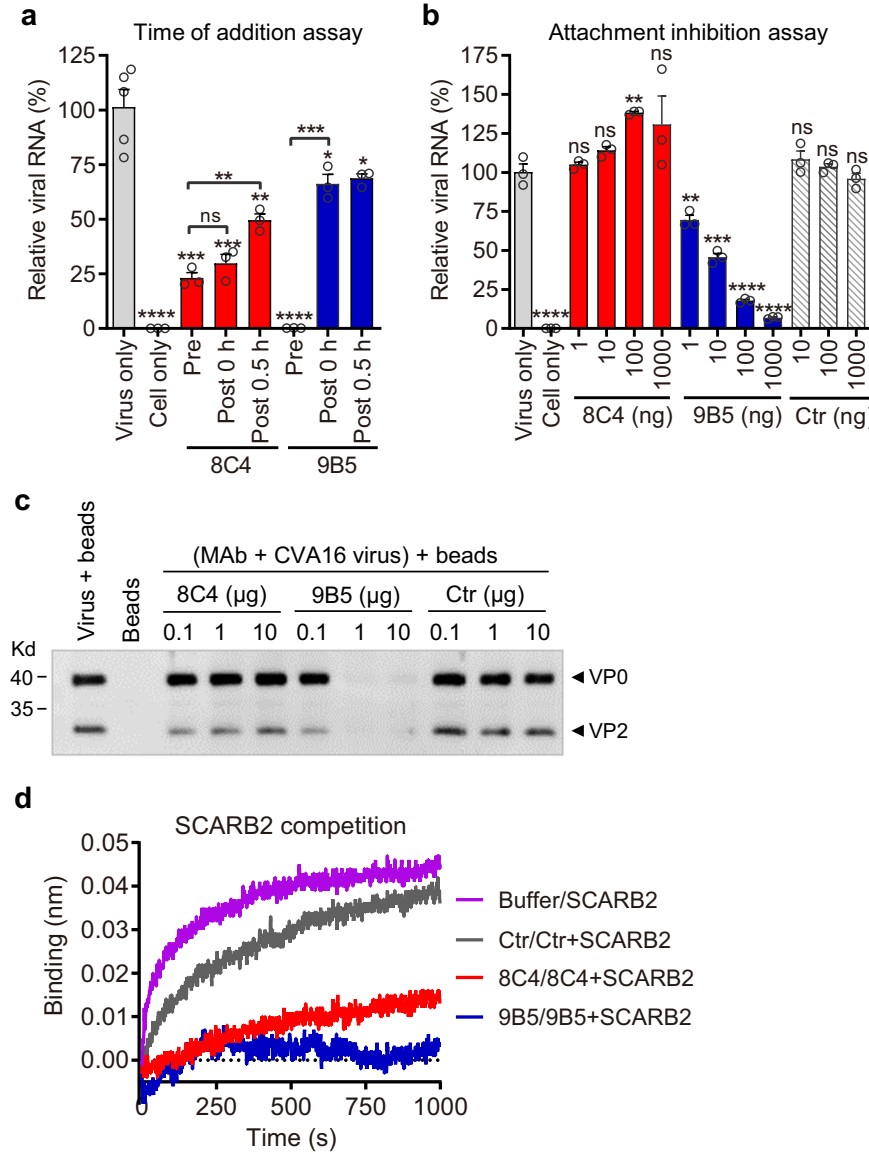

**Fig. 3 | Mechanisms of neutralization of the MAbs. a** Time of addition assay. CVA16/SZ05 was exposed to 8C4 or 9B5 before (Pre) or at different time points after (Post) the virus attached to pre-chilled RD cells. RNA was isolated at 6 h after infection for real-time RT-PCR analysis of CVA16 RNA. **b** Attachment inhibition assay. CVA16/SZ05 was incubated with various amounts of 8C4, 9B5, or control [Ctr] antibody 2H2 for 1 h, and the mixtures were cooled and then allowed to attach to pre-chilled RD cells for 1 h at 4 °C. After rinse, RNA was extracted for real-time RT-PCR analysis. In **a**, **b**, for each treatment, viral RNA levels relative to those for the only virus-infected samples are shown. Data are mean ± SEM of at least triplicate wells (*n* = 5 for the virus-only group in **a**, and *n* = 3 for each of the other groups). Each symbol represents one well of 24-well cell culture plate. Statistical significance between the virus-only and antibody-treated groups was calculated by two-tailed *t*-test. ns no significant difference (*p* ≥ 0.05); *\**p* < 0.05; *\*\**p* < 0.01; *\*\*\**p* < 0.001;

*\*\*\*\**p* < 0.0001. For **a**, *p* value between the 8C4-Pre group and the virus-only group is 0.0004; for the 8C4-Post-0 h group, *p* = 0.0007; for the 8C4-Post-0.5 h group, *p* = 0.0032; for the 9B5-Post-0 h group, *p* = 0.0197. For **b**, *p* value between the 8C4-100 ng group and the virus-only group is 0.0019; for the 9B5-1ng group, *p* = 0.0068; for the 9B5-10ng group, *p* = 0.0007. **c** Heparin-binding inhibition assay. CVA16/SZ05 was incubated with various amounts of 8C4, 9B5, or control antibody for 1 h, and the mixtures were then incubated with heparin-agarose beads for 1 h. The beads were collected, washed, and subjected to western blotting with anti-CVA16-VP0 antibody. **d** BLI-based SCARB2 competition binding assay. Immobilized CVA16/SZ05 viral particles were first incubated with buffer or the indicated MAb and then incubated with SCARB2-Fc protein in the presence of the MAb. The graphs show binding signals of SCARB2-Fc. For each panel, two independent experiments were performed, with similar results.

critical addition time point for 8C4 to mount neutralization and therefore 8C4 may block an early step at the post-attachment stage of viral entry. For 9B5, pre-treatment before viral attachment almost completely inhibited CVA16 infection (>99% inhibition), while only moderate inhibitory activity was observed when the antibody was added at 0 or 0.5 h post-infection (Fig. 3a), indicating that 9B5 exerts neutralization primarily at the pre-attachment stage of infection.

Next, we examined whether 8C4 or 9B5 could inhibit CVA16 attachment, the first step of the viral entry process. Briefly, CVA16 viral particles were pre-incubated with the MAbs before binding to cooled

RD cells at 4 °C, and after washing, RNA of CVA16 particles bound to cells was determined by quantitative RT-PCR. As shown in Fig. 3b, pre-treatment with MAb 8C4 or IgG control antibody had no inhibitory effect on viral binding, regardless of the tested antibody dose; in contrast, 9B5 pre-treatment could block the binding of CVA16 to RD cells in an antibody dose-related manner. These data indicate that 9B5, but not 8C4, can inhibit CVA16 attachment to host cells, in line with the results from the time-of-addition assay (Fig. 3a).

It has been previously shown that CVA16 binds soluble HS and utilizes cell surface HS glycosaminoglycans as its attachment

receptor[22]. We performed pulldown assays with heparin-conjugated agarose beads to determine whether 9B5 or 8C4 pre-treatment could block CVA16 binding to heparin. As shown in Fig. 3c, pre-treatment with 9B5 reduced the amounts of heparin-agarose bound CVA16 in an antibody dose-dependent manner, whereas neither 8C4 nor the control MAb affected CVA16 binding to heparin regardless of the antibody dose, indicating that 9B5 binding interferes with the CVA16-heparin interaction. Collectively, the above data demonstrate that 9B5 exerts neutralization primarily through blockade of CVA16 attachment to host cells via interfering with the interaction between the virus and its cellular attachment receptor heparin sulfate glycosaminoglycans.

Human SCARB2 has been previously reported to play a critical role in the uncoating of EV71 and CVA16[23,33]. To test whether pre-treatment with 8C4 or 9B5 could block CVA16 binding to SCARB2 receptor, we carried out a competition BLI assay with recombinant human SCARB2 luminal domain fused with human IgG Fc (SCARB2-Fc). Briefly, immobilized CVA16/SZ05 viral particles were pre-incubated with kinetics buffer (reference), the IgG control antibody 2H2[31], MAb 8C4, or 9B5 and then allowed to interact with SCARB2-Fc in the presence of the antibody. Binding signals of SCARB2-Fc were recorded and depicted in Fig. 3d. Compared with buffer alone, pre-incubation of control antibody with CVA16 merely led to a slight decrease in SCARB2-Fc binding signal. In contrast, pre-binding of 8C4 or 9B5 significantly or completely blocked subsequent SCARB2-Fc binding. These data suggest that 8C4 and 9B5 may interfere with CVA16 binding to cellular SCARB2 receptor located in the membranes of endosomes and lysosomes[34,35], therefore prevent receptor-mediated virus uncoating.

## Cryo-EM structure of CVA16 in complex with 9B5 Fab

To further investigate the structural basis of CVA16 neutralization by 9B5 MAb, we performed cryo-EM study on the CVA16–9B5 Fab complex. SDS-PAGE analysis of the purified CVA16 sample revealed the presence of VP0, VP1, VP2 and, VP3 capsid proteins, with VP0 being more abundant than VP2 (Supplementary Fig. 4), indicating that the CVA16 sample is a mixture of mature virions and empty particles. Inspection of the original micrographs revealed that both full and empty viral particles were highly decorated with Fabs (Supplementary Fig. 5a). From the same CVA16–9B5 dataset, we obtained three cryo-EM maps in distinct conformations namely C1, C2, and C3, at the resolution of 2.90, 3.35, and 3.80 Å and population distribution of 29.3%, 37.2%, and 33.5%, respectively (Fig. 4a–c and Supplementary Fig. 5). We then built an atomic model for each of the three CVA16–9B5 maps (Fig. 4d–f). Most of the side chain densities in the interaction interface between CVA16 and 9B5 Fab could be visualized (Fig. 4j).

Naturally occurring CVA16 viral particles are present in two distinct states, including the native/compact state (-161 Å in radius) and the expanded state (-170 Å)[16,17]. Here, our structures showed that the radii of the viral capsid of C1 and C2 are 161.4 Å and 162.3 Å, respectively, similar to that of the native CVA16 mature virion but smaller than that of C3 (169.2 Å) (Fig. 4a–c). The channels at the two-fold axes are closed in C1 and C2 maps; while that in C3 is open (Fig. 4a–f), which is a typic structural feature of expanded enteroviral particles[36]. Additionally, the VP1 hydrophobic pocket in C1 and C2 capsids is filled with a pocket factor, which functions to stabilize viral particles, but the VP1 pocket is empty and collapsed in the C3 capsid (Supplementary Fig. 6d–f). These features indicate that the C1 and C2 viral particles are in the native/compact state, while that in C3 is in the expanded state. Moreover, the central sections of the CVA16–9B5 maps revealed that the density corresponding to viral RNA genome is present inside the C1 capsid shell but absent in C2 and C3 capsids (Fig. 4g–i). Taken together, these data suggested: (1) the viral particle in C1 is in the native/compact state, similar to that of mature CVA16 virion; (2) the viral particle in C2 resembles the compact empty particle/procapsid; (3) the viral particle in C3 probably adopts the expanded empty particle

configuration. Our subsequent structural comparison showed similar architecture of the protomers between our C1 and the mature virion (overall Cα RMSD of 1.05 Å, Supplementary Table 3). The C2 is most similar to the compact empty particle (RMSD 1.16 Å), while C3 is similar to the expanded empty particle (RMSD 1.24 Å, Supplementary Table 3)[16,17]. This analysis substantiates the identities of the C1 to C3 viral particles.

In all of the CVA16–9B5 maps, 9B5 Fab binds to the five-fold vertex (Fig. 4a–c), with each CVA16 protomer bound with a Fab (Fig. 4k). The better resolved C1 in the native mature conformation was used to analyze the interaction interface between CVA16 and 9B5. Specifically, 9B5 Fab binds to the north rim of CVA16 canyon and partially obscures the canyon region (Fig. 4k), an important receptor-binding site for many EVs[37–39]. Both heavy and light chains of 9B5 Fab are involved in the antibody binding to CVA16 VP1 protein. Detailed analysis showed that the heavy chain complementarity-determining region 3 (HCDR3) and framework 3 (HFR3) of 9B5 bind to the BC loop (residues G99 and D104) of VP1, while the light chain CDR3 (LCDR3) contacts the VP1 EF (residue R166) and HI (residue K242) loops (Fig. 4l and Supplementary Table 4). NCBI BLAST analysis revealed that the four contacting residues, G99, D104, R166 and K242, in VP1 are highly or fully conserved among the analyzed 1127 CVA16 VP1 sequences with identity of 99.6%, 98.0%, 100%, and 99.9%, respectively (Supplementary Table 4). The CVA16–9B5 interaction interface covers a relatively small surface area (536.4 Å$^2$) on each protomer, and the 9B5 heavy and light chains contribute 51.3% and 48.7% of the overall binding interactions, respectively (Supplementary Table 5).

It is noted that residues R166 (EF loop) and K242 (HI loop) of VP1 could form salt bridges or hydrogen bond with residues D93 and W92 in LCDR3 of 9B5, respectively (Fig. 4l and Supplementary Table 4). It has been suggested that R166, K242, and three other positively charged residues (K141, K241, and H245) of VP1 are critical for CVA16 virion binding to the attachment receptor, HS[22]. Here our footprint analysis revealed that 9B5 binding could mask the five positively charged residues (Fig. 4m), thereby directly blocking the interaction between CVA16 virion and HS. This provides a structural explanation for the inhibition of CVA16 binding to HS in vitro by 9B5 antibody (Fig. 3c). In addition, SCARB2 was reported to be an important viral uncoating receptor for CVA16 and EV71[23,33]; however, the structure of the CVA16–SCARB2 complex has not been resolved yet. We then docked the structure of EV71–SCARB2 (PDB: 6I2K)[40] into our C1 structure (Fig. 4n). It appeared that despite 9B5 and SCARB2 bind to different regions of the viral capsid, the constant region of 9B5 clashes with the membrane proximal region of SCARB2 (bound to an adjacent protomer) in binding to CVA16 capsid (Fig. 4n). This explains why MAb 9B5 could inhibit virus binding to the soluble SCARB2 in vitro (Fig. 3d).

## Cryo-EM structure of CVA16 in complex with 8C4 Fab

To dissect the structural basis of CVA16 neutralization by 8C4, we determined the cryo-EM structure of CVA16 viral particles in complex with 8C4 Fab. From the same CVA16–8C4 dataset, we deduced three conformational states, namely C1, C2, and C3, at the resolution of 3.05, 2.91, and 3.36 Å and population distribution 15.5%, 27.0%, and 57.5%, respectively (Fig. 5a–c and Supplementary Fig. 7). We then built an atomic model for each of the three maps (Fig. 5d–f), and the side chain densities appear mostly well resolved in the CVA16–8C4 interaction interface (Fig. 5j).

For CVA16–8C4 C3, the radius of viral capsid is 170.5 Å, slightly larger than that of C1 and C2 (162.2 and 163.0 Å, respectively) (Fig. 5a–c), and the two-fold axis channel in its capsid is open, while that in C1 and C2 capsid is closed (Fig. 5a–f). Moreover, the VP1 hydrophobic pocket is empty and collapsed in C3 capsid, but filled with the pocket factor in C1 and C2 capsids (Supplementary Fig. 6g–i). Additionally, the viral RNA density is present in C1 but absent in C2 and C3 (Fig. 5g–i). Collectively, these data suggested that the viral particles

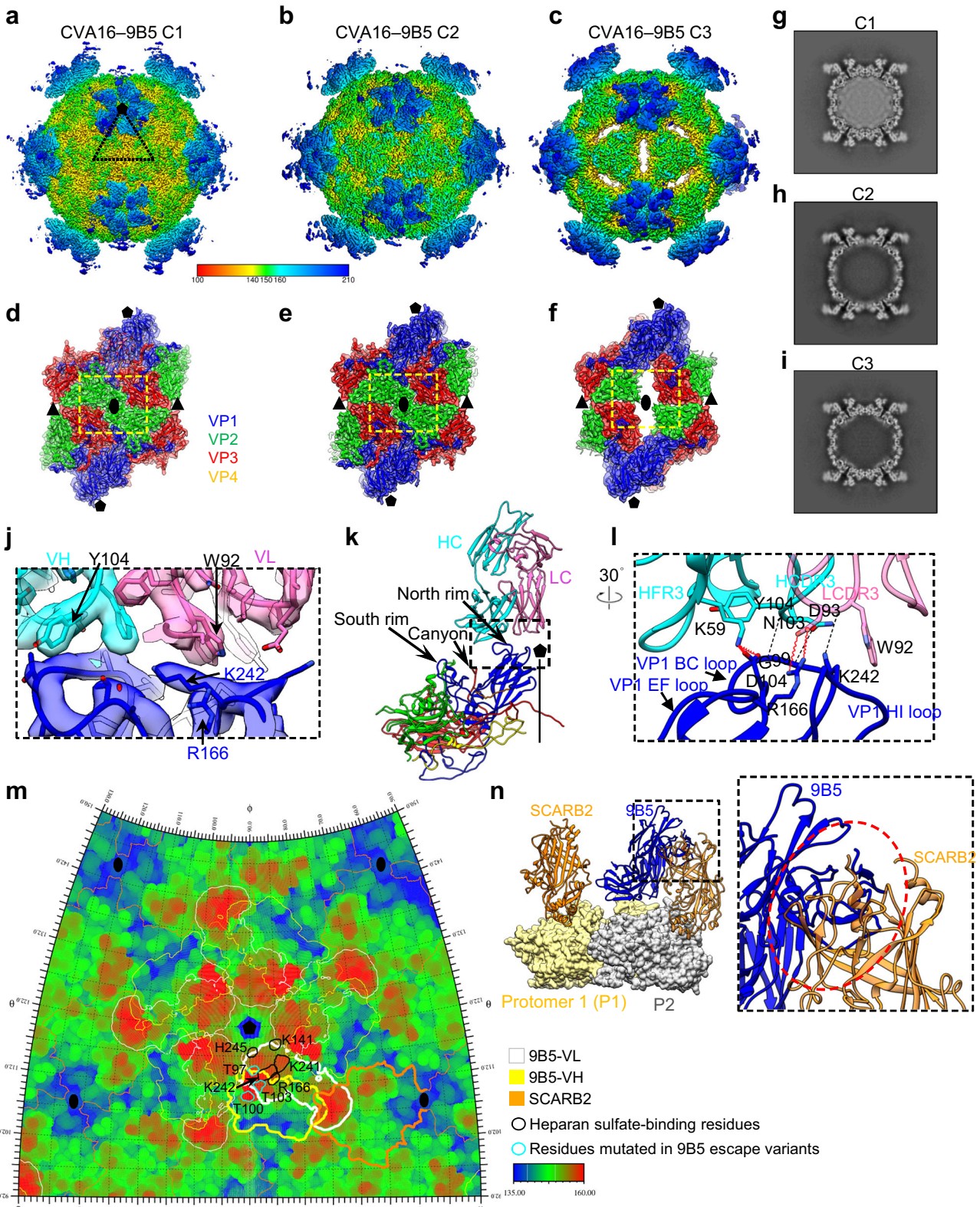

in C1, C2, and C3 adopt the native/compact mature CVA16 virion, the compact empty particle, and the expanded empty particle configuration, respectively. Moreover, structural comparison revealed that the protomers of the CVA16–8C4 C1/C2/C3 showed the most similar architecture to that of mature virion, compact empty particle, and expanded empty particle with the overall Cα RMSD values of 1.098, 1.178, and 0.937 Å, respectively (Supplementary Table 3), confirming the identity of these viral particles.

It appears that 8C4 Fab binds only to the native mature virion (C1) and compact empty particle (C2), rather than the expanded empty particle (C3) (Fig. 5a–c). Detailed analysis of the CVA16-8C4 C1 structure shows that 8C4 Fab binds closely to the icosahedral three-fold axis, and each Fab interacts with two adjacent protomers from different pentamers (Fig. 5j, k). Both heavy and light chains of 8C4 are involved in the engagement with VP3 from one protomer and VP2 from an adjacent protomer (denoted as VP2') (Fig. 5j, k and Supplementary

**Fig. 4 | Cryo-EM structures of CVA16 in complex with 9B5 Fab.** Cryo-EM maps of CVA16–9B5 in three distinct states, C1 (**a**), C2 (**b**), and C3 (**c**). The density maps are viewed along the two-fold axis. The color bar indicates the radius from the center of the particle (unit in Å). The black triangle indicates one icosahedral asymmetric unit. Density maps of the two-fold related protomers of the C1 (**d**), C2 (**e**), and C3 (**f**) conformers, superimposed with fitted models. 9B5 Fab was removed for clarity. VP1, VP2, VP3, and VP4 are colored in blue, green, red, and yellow, respectively; the same color scheme is used throughout. The black pentagon, ellipse, and triangle represent the five-fold, two-fold, and three-fold axes, respectively. The major differences among the conformers are indicated by yellow dashed rectangles. Central sections of the C1 (**g**), C2 (**h**), and C3 (**i**) density maps. **j** Zoomed-in view of the CVA16–9B5 C1 interaction interface, demonstrating that most side chain densities were well resolved. **k** Binding interface between CVA16 protomer and 9B5 Fab. VH and VL of 9B5 are colored in cyan and hotpink, respectively. The five-fold axis is also shown. **l** Zoomed-in views of the interactions between VP1 loops of CVA16 and the CDR and framework (FR) regions of 9B5. Black dashed lines indicate hydrogen bonds, and red springs indicate salt bridges. **m** Roadmap indicating the footprints of 9B5 on the CVA16 virion surface, obtained by RIVEM. Viral residues are colored by radius. VL and VH are indicated by white and yellow contour lines, respectively. The potential SCARB2-binding region is shown in orange contour lines. The positively charged viral residues (K141, R166, K241, K242, and H245) that bind heparan sulfate and the residues (T97, T100, and T103) mutated in 9B5 escape variants are indicated with black and cyan contour lines, respectively. **n** The structure of EV71–SCARB2 complex (PDB: 6I2K) was fitted into the CVA16 two adjacent protomers (P1 in khaki and P2 in gray), revealing that SCARB2 (orange) would clash with 9B5 Fab (blue) that is bound to the adjacent protomer.

Table 6). More specifically, the VP3 BC (residue S77) and HI (residue T210) loops engage with HCDR3 and LCDR1, respectively (Fig. 5l①); the VP2′ HI loop (residue G227, A228, S230, and E231) engages with LCDR3 and HCDR3 (Fig. 5l②); VP2′ βB (residue K69) and βC (residue W78) interact with HCDR2, while the VP2′ BC loop (residue D74) with HFR3 (Fig. 5l③); the VP2′ EF loop (residue V159) binds HCDR2 (Fig. 5l④). NCBI BLAST analysis showed that three contacting residues, K69, D74, and V159, in VP2 are extremely conserved (99.5–99.7%), while the other contacting residues in VP2 and VP3 are identical among all of the CVA16 strains analyzed (Supplementary Table 6). The CVA16–8C4 interaction interface buries a total surface area of 904.3 Å² on viral capsid, and the 8C4 heavy and light chains contribute 56.2% and 43.8% of the interface, respectively (Supplementary Table 5).

In the CVA16-8C4 C2 structure, the conformation of 8C4 binding site on the capsid is well maintained to accommodate the 8C4 Fab, despite both of the antibody and capsid protein display a very slight upward movement relative to the counterparts in the CVA16-8C4 C1 model (Supplementary Fig. 8a). Further analysis reveals that, compared to CVA16-8C4 C1, the C2 complex preserves most of the interactions between 8C4 and viral capsid (Supplementary Table 6). In contrast, compared with CVA16-8C4 C1, CVA16-8C4 C3 displays a quite large upward movement for the expanded capsid and in particular an obvious outward tilting movement of up to 5.2 and 4.7 Å toward the Fab for the protomer 1 and protomer 2, respectively (Supplementary Fig. 8b). Such a large movement toward the 8C4 Fab may create clashes between the CVA16 capsid and the 8C4 heavy chain, thus prevent 8C4 binding (Supplementary Fig. 8b). This potentially explained why 8C4 failed to bind CVA16 expanded empty particle.

Apparently, the 8C4 binding site resides on the three-fold protrusion of CVA16 viral particle and is far away from the binding site of the attachment receptor, HS, located around the five-fold axes of CVA16 viral capsid (Fig. 5a)[22], explaining why 8C4 binding does not block viral binding to HS in vitro (Fig. 3c). In addition, docking of the EV71–SCARB2 structure (PDB: 6I2K) into the CVA16–8C4 C1 structure revealed that the binding regions of 8C4 and SCARB2 on viral capsid are overlapping (e.g., the VP2 V159 residue), and both the variable and constant regions of 8C4 spatially clash with SCARB2 (Fig. 5m, n). Collectively, these data provide a structural explanation for the inhibition of CVA16 binding to SCARB2 protein in vitro by 8C4 antibody (Fig. 3d).

**The non-competing 8C4/9B5 cocktail prevents virus escape**

Consistent with the binding competition data (Fig. 1e–g), overlay of the cryo-EM maps of CVA16–9B5 C1 and CVA16–8C4 C1 showed that 8C4 and 9B5 recognize distinct, non-overlapping epitopes, thus may simultaneously bind the same CVA16 viral capsids (Fig. 6a). As a result, the two MAbs can be used as a non-competing antibody pair. The 8C4/9B5 cocktail was prepared by mixing the two antibodies at a 1:1 ratio and analyzed for neutralization against CVA16/SZ05 (Fig. 6b). The 8C4/9B5 cocktail showed slightly stronger neutralization activity than the 9B5 alone, indicating that 8C4 is compatible with 9B5 when used in

combination and the addition of 8C4 as the second component could further increase the neutralization strength against CVA16.

Next, we investigated the possibility of developing antibody-resistant virus mutants under the pressure of individual MAbs or the 8C4/9B5 cocktail. To screen escape mutants, CVA16/SZ05 was subjected to three consecutive passages in the presence of increasing concentrations of 8C4, 9B5, or the 8C4/9B5 cocktail. Representative data from one of two replicate experiments are shown in Fig. 6c–e. Mutants resistant to antibody 8C4 or 9B5 rapidly emerged at the second or third passage, whereas no mutants resistant to the 8C4/9B5 cocktail were isolated, indicating that the cocktail can prevent rapid mutational escape seen with individual MAbs. It is also observed that, although 9B5 alone is much more potent than 8C4 alone (neutralization concentration [100% protection] being 4.89 ng/ml vs. 1250 ng/ml), only 2.44 ng/ml of 9B5 in the 8C4/9B5 cocktail was needed to achieve the same level of neutralization. Altogether, the above data indicate the 8C4/9B5 cocktail is superior to individual MAbs in rendering neutralization and in preventing virus escape.

**Viral mutations responsible for MAb resistance**

The selected mutant viruses were then subjected to plaque purification and sequencing of the capsid protein-coding region (Fig. 6f). 8C4-resistant mutants were isolated from two different wells (the upper and lower wells were designated well #1 and #2, respectively) of the culture plates (Fig. 6c). In plaque purification experiments, we found that 8C4-resistant mutants from well #1 formed plaques at 1/10² to 1/10³ dilutions (the lowest dilution tested was 1/10²), while the mutants from well #2 at 1/10² to 1/10⁵ dilutions. Thus, 8C4-resistant mutants from well #1 had lower fitness than those from well #2 and wild-type CVA16. Sequence analysis of escape mutants showed that all (8/8 plaques) of the 8C4-selected isolates from well #1 harbored a V159F mutation in VP2, while those from well #2 carried a K69Q mutation in VP2 (Fig. 6f). These results are consistent with structural observations that VP2 V159 and K69 are directly involved in binding to HCDR2 of 8C4 (Fig. 5l③④ and Supplementary Table 6). According to the crystal structure of CVA16 mature virion (PDB: 5C4W)[16], VP2 V159 and K69 residues are located in VP2 EF loop and βB, respectively, and are exposed at the surface of viral capsid (Fig. 6g). Both VP2 V159 and K69 residues are extremely conserved (99.5% and 99.7% identity, respectively) among CVA16 strains (Supplementary Table 6), suggesting that they may be essential to CVA16 infectivity. Moreover, the 8C4-selected escape mutations V159F and K69Q are not present in all of the currently circulating CVA16 strains. VP2 V159 is also predicted to interact with SCARB2 receptor (Fig. 5n), and mutations at this position may affect the SCARB2-binding affinity of CVA16 viral particles, thus explaining the substantial reduction in viral fitness caused by the VP2 V159F mutation (Fig. 6f). The 9B5-resistant mutants formed plaques at 1/10² to 1/10⁵ dilutions and contained the T97A + T100A (2/8 plaques) or T100A + T103A (6/8 plaques) double mutations in VP1 (Fig. 6f). VP1 T97, T100, and T103 residues are situated in VP1 BC loop and exposed at the virion surface (Fig. 6g). NCBI BLAST analysis showed that VP1

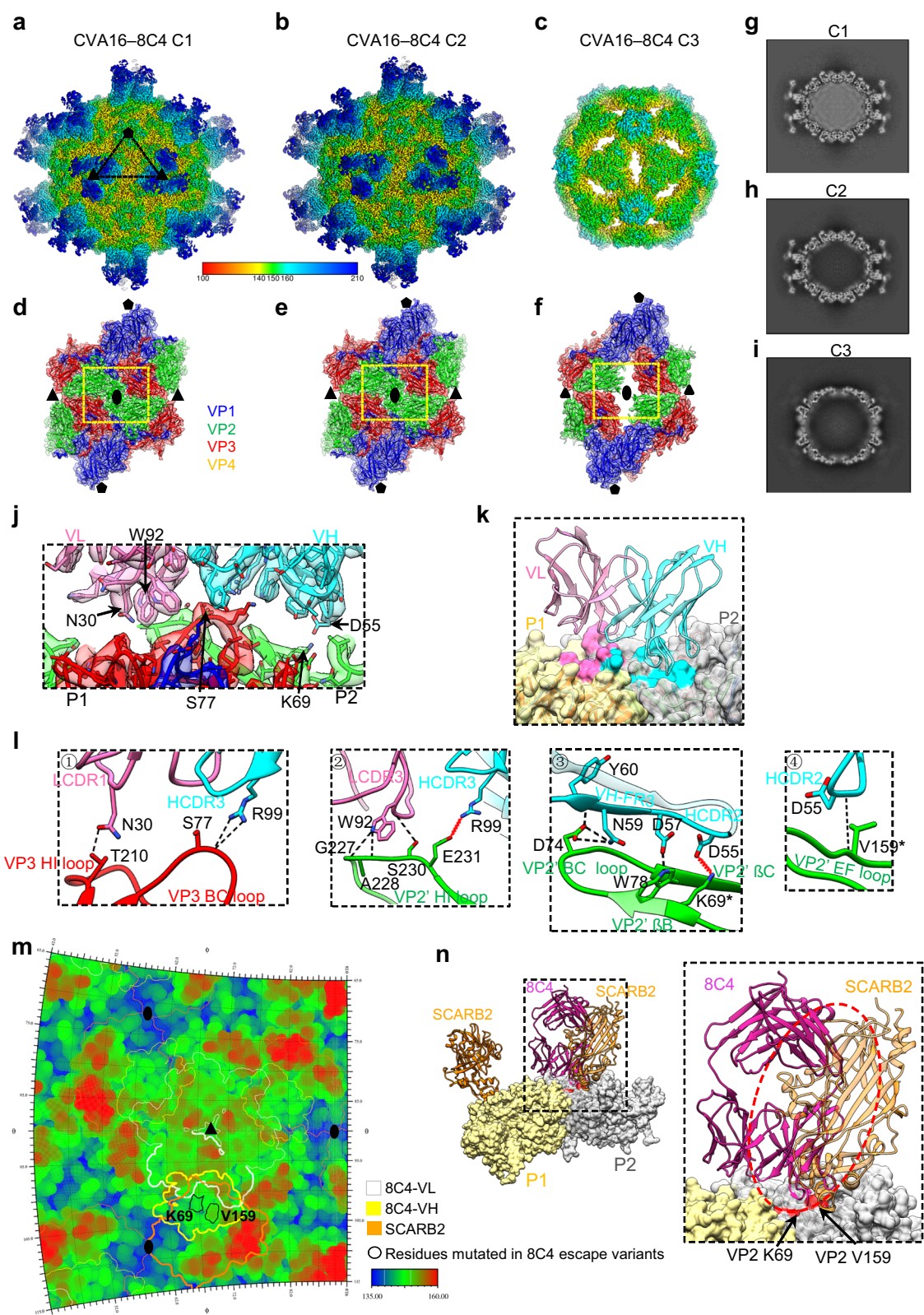

T97, T100, and T103 are highly conserved across all CVA16 strains with conservation of 99.5%, 99.9%, and 98.2%, respectively. Moreover, the 9B5-selected escape mutations T97A + T100A or T100A + T103A are not present in all of the currently circulating CVA16 strains. In addition, structure analyses of the CVA16–9B5 complex show that VP1 BC loop forms direct contacts with 9B5 Fab (Fig. 4l), and VP1 T97, T100, and T103 residues are indeed within the footprint of 9B5 (Fig. 4m). Thus,

the findings from the antibody-escaping mutant analysis are in well agreement with the results of our structural studies.

## Discussion

Neutralizing antibodies play an important role in anti-viral immunity[26], therefore thorough understanding of their working mechanisms may inform and facilitate the design of optimal anti-viral strategies.

**Fig. 5 | Cryo-EM structures of CVA16 in complex with 8C4 Fab.** Cryo-EM density maps of CVA16–8C4, resolved in three different conformations, namely C1 (**a**), C2 (**b**), and C3 (**c**). Density maps of the two-fold related protomers of the C1 (**d**), C2 (**e**), and C3 (**f**) conformers, superimposed with fitted models. 8C4 Fab was removed for clarity. Central sections of the C1 (**g**), C2 (**h**), and C3 (**i**) density maps. **j** Enlarged view of the CVA16–8C4 C1 interaction interface, demonstrating that the side chain densities of the key residues were well resolved. Each 8C4 Fab interacts with two adjacent protomers, P1/P2, from different pentamers. **k** The interaction interface between 8C4 Fab (ribbon diagram) and two adjacent CVA16 protomers (surface). Footprints of VH and VL of 8C4 on the capsid are colored with cyan and hotpink, respectively. **l** Enlarged views of the interactions between VP3 (protomer 1) BC, HI loops and VP2' (protomer 2) BC, EF, HI loops, βB, βC of CVA16 and the CDR and FR regions of 8C4. **m** Roadmap showing the footprints of 8C4 Fab on the CVA16 virion surface. The VL, VH of 8C4 Fab and the potential SCARB2 are indicated by white, yellow and orange contour lines, respectively. The residues (K69 and V159) mutated in the 8C4 escape mutants are indicated with black contour lines. **n** The EV71–SCARB2 structure (PDB: 6I2K) was fitted into the CVA16 two adjacent protomers, indicating that SCARB2 (orange) that is bound to P2 would clash with 8C4 Fab (violet red).

Although some anti-CVA16 neutralizing MAbs have been identified[17,41] and several of them have been structurally characterized[17], their mode of action and inhibitory mechanisms remain elusive. In this study, we developed two neutralizing anti-CVA16 MAbs, namely 9B5 and 8C4, and used them to probe molecular mechanisms of antibody-mediated neutralization of CVA16. Interestingly, we found that the two antibodies adopt different mechanisms of action, and the main differences are summarized in Supplementary Fig. 9a. Essentially, 9B5, which binds around the five-fold apex of the capsid, (1) mounts neutralization primarily at pre-attachment stage of viral entry, (2) potently inhibits virus attachment to susceptible cells through blocking the interaction between CVA16 viral particles and the attachment receptor HS, and (3) is also able to interfere with CVA16 interaction with its uncoating receptor SCARB2 (Fig. 3). For 8C4 that targets the three-fold protrusion, it (1) exerts inhibitory function mainly at an early post-attachment step of viral entry, (2) cannot block virus attachment to cells and virus binding of HS, and (3) is able to block the interaction between CVA16 and the uncoating receptor SCARB2 (Fig. 3). Our structural studies reveal that 9B5 binding largely obscures the HS binding site and also creates steric hindrance which potentially blocks SCARB2 binding to the CVA16 particle (Fig. 4m, n), whereas the 8C4 binding epitope is far away from the HS binding site but overlaps with the SCARB2 binding site on the viral capsid (Fig. 5). Thus, our experimental evidences are in well agreement with our structural observations. Together, these findings elucidate for the first time to our knowledge the detailed working mechanisms of anti-CVA16 neutralizing antibodies.

Our neutralizing MAbs 9B5 and 8C4 define two neutralization sites located around the five-fold and three-fold axes, respectively. A previous study has also reported three CVA16-specific neutralizing MAbs (18A7, NA9D7, and 14B10) that target distinct epitopes[17]. Based on the antibody binding sites, these five neutralizing MAbs can be divided into three groups: group 1 consists of 9B5 and 18A7, both of which target the five-fold protrusion of CVA16 viral particle; group 2 contains only NA9D7, whose epitope is located around the two-fold axis; group 3 is comprised of 8C4 and 14B10, both of which bind CVA16 capsid around the three-fold axis (Supplementary Fig. 9b). The binding footprints and the areas of contact of the three groups of anti-CVA16 antibodies are summarized in Supplementary Fig. 9c, d. Our in vivo protection study demonstrated that 9B5 and 8C4 are highly effective in both prophylactic and therapeutic settings (Fig. 2). Whereas the antibodies NA9D7 and 14B10, but not 18A7, were able to confer protection in the mouse model, despite the neutralizing activity of 18A7 (IC50 = 40 ng/ml) is much stronger than that of NA9D7 (IC50 = 1960 ng/ml) and 14B10 (IC50 = 1010 ng/ml)[17]. It is intriguing that 9B5 and 18A7, both belonging to the five-fold vertex-binding antibody group 1, displayed drastically different efficacies in vivo. To understand the mechanism underlying this considerable difference, we carefully compared the binding modes on mature virion between the two antibodies. The binding footprint of 18A7 is much smaller than that of 9B5 (Supplementary Fig. 9c, d). Only a single 18A7 Fab molecule binds to one five-fold vertex of viral capsid nearly along the five-fold axis (Supplementary Fig. 9e), thus 12 copies of 18A7 Fab are attached to each icosahedral CVA16 particle[17]. For 9B5, five Fab molecules engage each five-fold vertex at an angle of ~43.9° to the five-fold axis and thus

totally 60 copies of 9B5 Fab bind to each viral particle (Supplementary Fig. 9d, f). Compared with 18A7, 9B5 exhibits a more oblique binding angle (more tilted toward the canyon region) and larger covering area on the viral surface (Supplementary Fig. 9f), which may lead to more efficient blockade of the interactions between the virus and its cellular receptor(s) both in vitro and in vivo. The comparison of 9B5 and 18A7 suggests that antibody occupancy on the mature virion is a key determinant of the protective efficacy for the group 1 antibodies. Our study demonstrates that, besides the two-fold or three-fold vertex-targeting antibodies (e.g., NA9D7, 8C4, and 14B10), neutralizing antibodies (such as 9B5 or 9B5-like antibodies) that bind to the five-fold vertex of CVA16 capsids can also be protective in vivo, thus revealing a more complete anti-CVA16 protective epitope landscape on the CVA16 capsid.

Most structural studies on EV71 or CVA16 show that naturally occurring EV71 or CVA16 particles exist in two forms: mature virion in the compact state and empty particle in the expanded state[17,18,42]. However, a previous study on CVA16 indicated that formalin-treated CVA16 empty particle may exist in compact state[16]. In the current study, we found that the CVA16–9B5 C2 structure has a capsid radius (162.3 Å) similar to that (161.4 Å) of the compact mature virion in the C1 structure but does not contain viral RNA genome inside the capsid, and also presents closed, but not open, two-fold channels (Fig. 4) (note that closed channel at the two-fold axis is a typical structural feature of compact enteroviral particles[16,36]), indicating that the viral particle in the CVA16–9B5 C2 structure is indeed compact empty particle. A similar conclusion can also be drawn based on our CVA16–8C4 C2 map. Collectively, these data demonstrate the existence of naturally occurring compact empty particle of CVA16. Interestingly, we found that 9B5 and 8C4 exhibit different binding patterns to the three forms of CVA16 particles, including compact mature virion, compact empty particle, and expanded empty particle. Specifically, 9B5 is able to bind all three particle forms, whereas 8C4 recognizes only the compact particles (including mature virion and compact empty particle) but not the expanded empty particle (Figs. 4 and 5). These data show that, similar to the CVA16 compact mature virion, the CVA16 compact empty particle well maintains major protective epitopes on its capsid surface and is therefore a good candidate for vaccine development. Our findings suggest a revision to the previous notion that the mature virion of CVA16 should be targeted for vaccine design[17]. We propose that both CVA16 mature virion and compact empty particle should be considered in designing and producing anti-CVA16 vaccines with optimal efficacy.

Picornaviruses are known to have high nucleotide substitution rates and evolve very rapidly among viruses[43], possibly leading to emergence of escape mutants under antibody selective pressure. Indeed, we found that although neutralization potency of MAbs 9B5 and 8C4 was very high, neutralization-escape mutants could be rapidly generated (observed as early as the second passage) under selective pressure of individual MAb (9B5 or 8C4 alone) (Fig. 6). However, in the presence of both 9B5 and 8C4, no escape mutant was detected (Fig. 6), demonstrating that the 9B5/8C4 cocktail can effectively circumvent the generation of viral escape mutants. Antibodies 9B5 and 8C4 complement each other, because they target non-overlapping

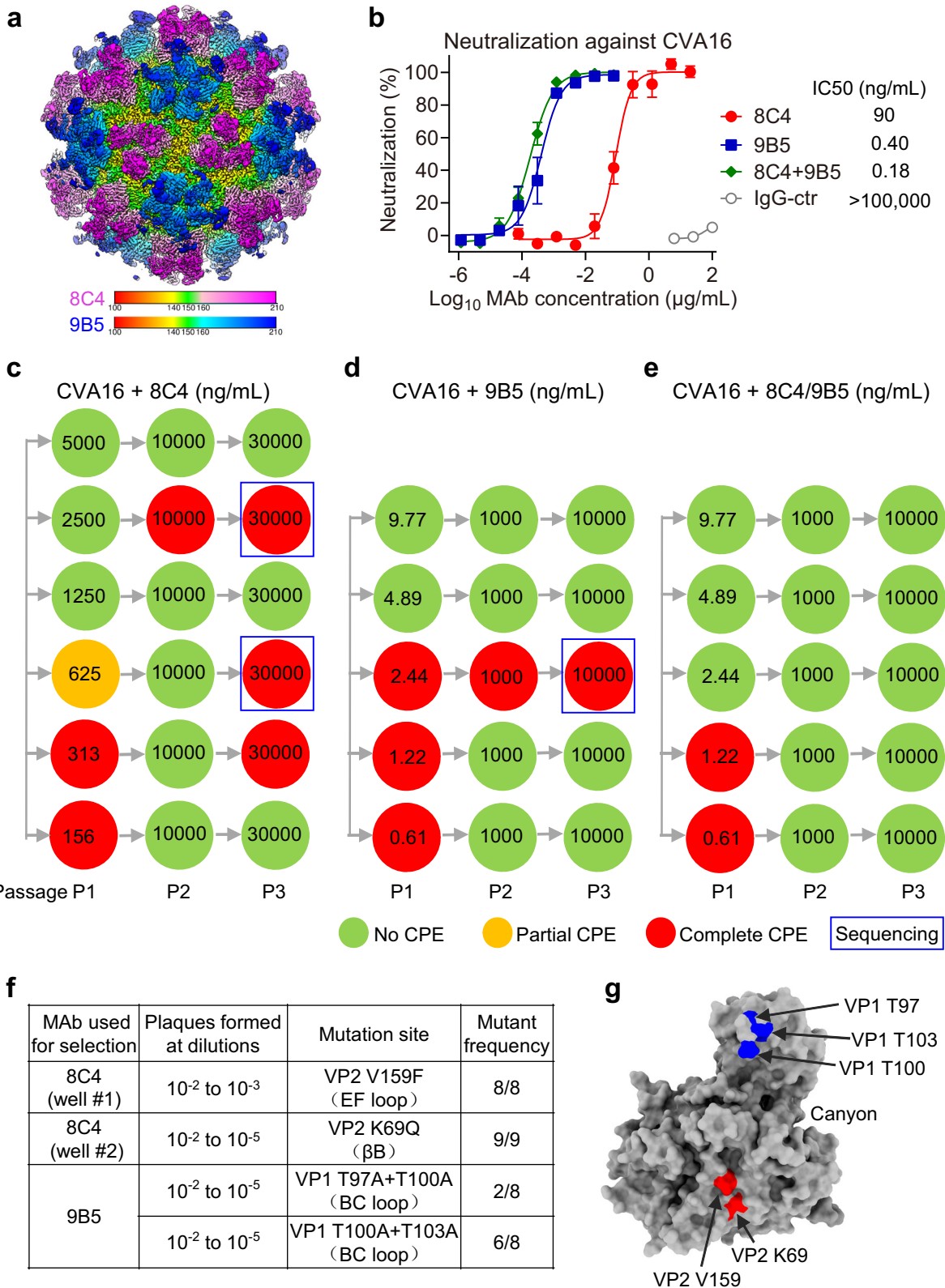

neutralization sites and adopt distinct modes and mechanisms of neutralization (Supplementary Fig. 9a), and their combination displays improved potency and unique ability to prevent virus escape. Therefore, further development of the 9B5/8C4 antibody cocktail into an anti-CVA16 therapy is warranted. In addition, our findings also suggest that mature virion- or compact empty particle-based vaccines will unlikely induce the generation of viral escape mutants, because of the

polyclonal nature of vaccine-elicited antibodies that target multiple neutralization sites.

In summary, our study clearly defines the function and working mechanism of two groups of CVA16-specific neutralizing MAbs that target the five-fold and three-fold vertexes of the capsid, and reveals the existence of naturally occurring CVA16 compact empty particle that highly resembles the CVA16 mature virion in overall capsid

**Fig. 6 | A combination of MAbs 8C4 and 9B5 prevents CVA16 virus escape mutations. a** Overlay of the cryo-EM maps of CVA16–9B5 C1 and CVA16–8C4 C1. The color bars indicate the corresponding radius from the center of the particle (unit in Å). The densities for 8C4 and 9B5 Fabs were colored in violet red and blue, respectively. **b** Neutralization of single MAbs and the antibody cocktail against CVA16/SZ05 was measured by the cell viability assay. 8C4 + 9B5, MAbs 8C4 and 9B5 were combined at a ratio of 1:1. For MAb cocktail, the concentration on the $x$ axis is that of the 9B5 antibody and only IC50 value for 9B5 is shown. IgG-ctr, SARS-CoV-2 MAb 2H2. Data are mean ± SEM of five replicate wells. **c–e** Screening of neutralization escape mutants. 100 $TCID_{50}$ of wild-type CVA16/SZ05 was incubated with various concentrations of MAb 8C4 (**c**), 9B5 (**d**), or 8C4 + 9B5 cocktail (**e**) for 1 h before infection of RD cells. After 4 days, the culture supernatants and cell lysates

(passage 1 [P1] virus) were incubated with the indicated concentrations of the MAbs prior to infection of fresh RD cells. After 3 days, the resultant P2 virus was subjected to another round of selection with higher concentrations of the MAbs, yielding P3 virus. During these passages, cells were observed for CPE. Green circle indicates no CPE, orange circle indicates partial CPE, and red complete CPE. Antibody concentrations used are shown in the circles. For the cocktail, only the concentrations of MAb 9B5 are shown. The mutants used for sequencing were blue boxed. **f** Information of neutralization escape mutants selected by MAb 8C4 and 9B5. **g** Location of escape mutations on the CVA16 protomer (PDB: 5C4W) surface. The residues replaced in 8C4-selected and 9B5-selected mutants are colored in red and blue, respectively.

structure and antigenic property. Our work also demonstrates the 9B5/8C4 antibody cocktail displays increased neutralization potency and acquires a unique ability to prevent virus escape seen with individual antibodies. These findings should have important implications for design and development of anti-CVA16 vaccines and therapies.

# Methods

## Ethics
The mouse studies were approved by the Institutional Animal Care and Use Committee at the Institut Pasteur of Shanghai. The mice were kept in the specific pathogen free animal facility with controlled temperature (20–26 °C), humidity (40–70%), and lighting conditions (12 h light/12 h dark cycle).

## Cells and viruses
Vero and rhabdomyosarcoma (RD) cells were grown in DMEM (Gibco, USA) with 5% fetal bovine serum (FBS). Mouse SP2/0 cells were grown in RPMI 1640 medium (Gibco) with 10% FBS. CVA16 strains used in this study include CVA16/SZ05 and a mouse-adapted strain CVA16/MAV[28]. EV71 strain EV71/G082 has been described previously[44]. All viruses were titrated by the 50% tissue culture infectious dose ($TCID_{50}$) assay.

## Proteins and antibodies
To obtain CVA16 viral particles, Vero cells were grown to 80% confluency and then infected with CVA16/SZ05 at a multiplicity of infection of 0.01. At 2 days post-infection (dpi), culture supernatant was collected, and the cells were lysed in 0.15 M PBS buffer with 1% NP-40 and then clarified by centrifugation. Next, the culture supernatant and the clarified cell lysate were precipitated with 10% polyethylene glycol (PEG) 8000 and 200 mM NaCl. After high-speed centrifugation, the pellet was collected and resuspended in 0.15 M PBS buffer. After centrifugation to remove insoluble material, the supernatant was subjected to 20% sucrose cushion ultra-centrifugations at 112,700 × $g$ for 5 h. The resulting pellet was resuspended in 0.15 M PBS buffer and centrifuged to remove insoluble material. Next, the supernatant was further purified by 10–50% sucrose gradient ultra-centrifugations at 270,000 × $g$ for 3 h. The fractions containing virus antigens were pooled and concentrated using Amicon Ultra 100 K filters (Millipore, USA). The final CVA16 virion sample was analyzed by SDS-PAGE and quantified by Bradford assay.

To generate SCARB2 protein, DNA fragment encoding the luminal domain of human SCARB2 (residues V28 to L433) was cloned into a modified pcDNA3.4 vector with an N-terminal IL-10 signal sequence and C-terminal human IgG1 Fc and His tag using ClonExpress II One Step Cloning Kit (Vazyme, China), yielding plasmid pcDNA3.4-SCARB2-hFc. The plasmid was then transfected into HEK 293F suspension cells using Polyethylenimine (PEI; PolySciences, USA). After 5 days of culture, the culture supernatant was harvested by centrifugation and subjected to affinity purification using Ni-NTA resin (Millipore, USA) to obtain His-tagged SCARB2-hFc fusion protein.

MAb 2H2 is an IgG antibody against SARS-CoV-2[31] and used as a negative control in this study. MAb D5 is an antibody against EV71[44].

## Preparation and sequencing of MAbs and Fabs
Adult female BALB/c mice (inbred strain, Vital River Laboratory Animal Technology company, China) were immunized intraperitoneally (i.p.) three times at 2-week intervals with 4 µg/dose of inactivated CVA16 antigen[28] in aluminum adjuvant (Thermo Fisher Scientific, USA). About 2 weeks after the third immunization, one mouse was injected intravenously with 10 µg of inactivated CVA16 antigen[28] in PBS. Three days after the boost, splenocytes were taken from the mouse and fused with SP2/0 myeloma cells using PEG 1450 (Sigma, USA), followed by HAT (Sigma) selection for 10 days. Next, hybridoma supernatants were screened by neutralization assay with CVA16/SZ05 as described below. Positive hybridomas were cloned 2–4 times to generate monoclonal stable cell lines. Antibody isotypes were measured using SBA Clonotyping™ System/HRP ELISA kit (Southern Biotech, USA) according to manufacturer's protocol. Heavy and light chain variable region sequences were determined using the 5′ RACE System (Invitrogen, USA) or mouse Ig-primer set (Novagen, Merck, Germany) according to the manufacturer's protocols and then analyzed using the IgBLAST tool[45].

MAbs were purified from ascites using protein G agarose resin 4FF (Yeasen, China) according to our previously described protocol[46]. Fabs were obtained by papain digestion of anti-CVA16 MAbs and then purified using Protein G column and Sephacryl S-100 column (GE Healthcare, USA) according to our previously described protocols[47].

## Neutralization assay
Neutralizing activities of hybridoma supernatants and purified MAbs against CVA16 or EV71 were determined by micro-neutralization assay. Briefly, 50 µl/well of undiluted hybridoma culture supernatants or serially diluted purified MAbs were mixed with 50 µl/well (100 $TCID_{50}$) of CVA16/SZ05, CVA16/GX08, CVA16/MAV, or EV71/G082 in 96-well plates, followed by incubation at 37 °C for 1 h. Next, 100 µl/well (20,000 cells) of RD cells were added to the plates and incubated at 37 °C. After about 3 days, the cultures were observed for CPE, and cell viability was then measured using CellTiter-Glo 2.0 assay kit (Promega, USA) according to the manufacturer's instructions. Luminescence measurements are expressed as relative luminescence units (RLU). Percent neutralization was calculated by the following equation: 100 × (RLU of the sample − RLU of the virus control wells) / (RLU of untreated cells − RLU of the virus control wells). For each MAb, its IC50 (half inhibitory concentration) was calculated using GraphPad Prism software by nonlinear regression.

## Bio-layer interferometry (BLI) assay
Before BLI assay, purified CVA16/SZ05 viral particles were labeled with EZ-Link™ Sulfo-NHS-LC-LC-Biotin (Thermo Fisher Scientific) and then purified using Zeba™ spin desalting column (Thermo Fisher Scientific) according to the manufacturer's protocols. The virus-binding affinity of the MAbs was determined by BLI assay in an Octet® RED96 System (Pall FortéBio, USA). Briefly, biotinylated CVA16/SZ05 viral particles were immobilized onto streptavidin biosensors (Pall FortéBio). Next, these biosensors were transferred into wells containing a series of

diluted individual antibody samples to permit virus-antibody association and then transferred into dissociation buffer (0.01 M PBS with 0.1% bovine serum albumin and 0.02% Tween 20). Equilibrium dissociation constants (KD) were calculated using Octet data analysis software v11.0 (Pall FortéBio).

For antibody competition assay, CVA16/SZ05 viral particles-coated sensors were transferred into wells containing buffer (control) or 15 μg/ml of the first MAb (the competitor antibody) for 500 s. The sensors were then moved into wells containing the second MAb alone (15 μg/ml; reference) or the antibody mixture (15 μg/ml of the first MAb plus 15 μg/ml of the second MAb) for 500 s. Octet data analysis software was used to calculate the binding level of the second MAb.

## In vivo protection assays

The protective efficacy of anti-CVA16 MAbs was assessed in a mouse model of CVA16 infection based on the mouse-adapted strain CVA16/MAV[28]. Pregnant female Institute of Cancer Research (ICR) mice (outbred model) were purchased from Vital River Laboratory Animal Technology company.

For the prophylactic experiment, groups of naïve 1-day old ICR mice were i.p. injected with PBS, 10 mg/kg of anti-CVA16 MAbs (8C4 or 9B5), a mixture of both 8C4 and 9B5 (10 μg/g of each MAb), or 10 mg/kg of control MAb 2H2[31]. After 24 h, the mice were i.p. infected with 1875 $TCID_{50}$ of CVA16/MAV strain.

For the therapeutic assay, groups of 2-day-old ICR mice were i.p. injected with 1875 $TCID_{50}$ of CVA16/MAV strain. After 24 h, the mice were i.p. injected with PBS, 10 mg/kg of MAb 8C4, 10 mg/kg of MAb 9B5, or a mixture of both 8C4 and 9B5 (10 mg/kg of each MAb). For the prophylactic and therapeutic assays, all infected mice were observed daily for survival and clinical score for 14 days. Clinical scores were graded as follows: 0, healthy; 1, reduced mobility; 2, limb weakness; 3, limb paralysis; 4, death.

## Time of addition assay

Time of addition assay was performed according to our previously described protocol[48]. Briefly, for pre-attachment inhibition assay, 1000 $TCID_{50}$ of CVA16/SZ05 was mixed with 1 μg of each of the MAbs, and the virus-antibody mixtures were cooled on ice and then added to pre-chilled RD cells cultured in 24-well plates, followed by incubation at 4 °C for 1 h to permit viral attachment. The cells were then washed twice with chilled PBS, and fresh DMEM with 1% FBS was added and incubated at 37 °C.

For post-attachment inhibition assay, 1000 $TCID_{50}$ of CVA16/SZ05 was added to pre-chilled RD cells and allowed to adsorb for 1 h at 4 °C. Next, the cells were washed twice with cold PBS and then incubated with fresh medium at 37 °C for 0 or 0.5 h to permit viral entry. Next, fresh media supplemented with 1 μg of individual MAbs was added to the wells, and the cells were re-incubated at 37 °C. For both assays, culture supernatant and RD cells were harvested together 6 h after infection and subjected to RNA isolation using TRIzol reagent (Invitrogen). cDNA was synthesized from the RNA using PrimeScript RT reagent Kit (Takara, Japan) and then subjected to real-time PCR analysis using SYBR Premix Ex Taq kit (Takara) with LightCycler® 480 II system (Roche, Switzerland) according to the manufacturer's protocols. CVA16-specific primers were as follows: forward primer, 5′-ATCCAGTAAGGATCCCAGACT-3′; reverse primer, 5′-GATTTGCATAGTGGAGAGCAG-3′. β-actin primers were as follows: forward primer, 5′-GGACTTCGAGCAAGAGATGG-3′; reverse primer, 5′-AGCACTGTGTTGGCGTACAG-3′. Data were analyzed using the $2^{-\Delta\Delta Ct}$ method with β-actin as the internal control.

## Inhibition of virus attachment by the MAbs

In total, $1.0 \times 10^6$ $TCID_{50}$ of CVA16/SZ05 was incubated with various amounts (1, 10, 100, or 1000 ng) of each of the MAbs at 37 °C for 1 h. The mixtures were cooled on ice, then added to pre-chilled RD cells in

24-well plates and allowed to adsorb for 1 h at 4 °C. The cells were washed twice with cold PBS and then subjected to RNA isolation using TRNzol Universal reagent (TIANGEN, China). cDNA was synthesized and subjected to real-time PCR analysis with CVA16-specific and β-actin-specific primers as described above. Data were analyzed using the $2^{-\Delta\Delta Ct}$ method with β-actin as the internal control.

## Heparin pulldown assay

In total, $1.0 \times 10^7$ $TCID_{50}$ (500 μl) of CVA16/SZ05 was incubated with various amounts (0.1, 1, or 10 μg) of each of the MAbs at room temperature for 1 h. The virus-MAb mixtures were then mixed with 20 μl of heparin-agarose beads (Sigma), followed by incubation at room temperature for 1 h under gentle rotation. The beads were collected by centrifugation at $835 \times g$ for 3 min, washed with PBS twice and resuspended in SDS loading buffer, followed by western blotting analysis with rabbit anti-CVA16-VP0 polyclonal antibody as primary antibody (1:1000 dilution) and goat anti-rabbit IgG−HRP (sigma) as secondary antibody (1:10,000 dilution).

## BLI-based SCARB2 competition assay

Biotinylated CVA16/SZ05 viral particles were coated onto streptavidin biosensors. Next, these biosensors were moved into wells containing buffer (control) or 15 μg/ml of each of the MAbs for 500 s. The sensors were then moved into wells containing SCARB2-hFc protein alone (180 μg/ml; reference) or the MAb and SCARB2-hFc mixture (15 μg/ml of MAb plus 180 μg/ml of SCARB2-hFc) for 1000 s. Octet data analysis software was used to calculate the binding level of SCARB2-hFc.

## Cryo-EM sample preparation

To prepare immune complexes, CVA16 virion and 9B5 or 8C4 Fab were incubated at a molar ratio of 1:120 for 20 min at room temperature. An aliquot of 3 μl of CVA16−9B5 or CVA16−8C4 complex was placed onto a plasma-cleaned holey carbon grids (R2/1, 200 mesh; Quantifoil Micro Tools) or a continuous ultrathin carbon film covered lacey carbon grid (400 mesh; Ted Pella), respectively. The grids were blotted and plunged into liquid nitrogen-cooled liquid ethane with a Mark IV Vitrobot (Thermo Fisher Scientific).

## Cryo-EM data collection

Cryo-EM movies of the samples were collected on a Titan Krios transmission electron microscope (Thermo Fisher Scientific) operated at an accelerating voltage of 300 kV, using FEI TEM user interface 2.15.3. The movies were recorded using a K3 Summit direct electron detector (Gatan) in counting mode (yielding a pixel size of 1.1), in an automatic manner using EPU 2.11 software (Thermo Fisher Scientific). For CVA16−9B5 complex, each frame was exposed for 0.05 s, and the total exposure time was 3 s, leading to a total accumulated dose of -38 e⁻/Å² on the specimen. For CVA16−8C4 complex, each frame was exposed for 0.1 s, and the total exposure time was 3 s, leading to a total accumulated dose of -50 e⁻/Å². Defocus values for both complexes ranged from −0.5 to −2.0 μm (Supplementary Table 2).

## Cryo-EM single particle 3D reconstruction

For each dataset, the motion correction of image stack was performed using the embedded module of Motioncor2 in Relion 3.1[49,50], and CTF parameters were determined using CTFFIND4.1.8[51] before further data processing. Unless otherwise described, the data processing was mainly performed in Relion 3.1[50].

For the CVA16−9B5 dataset (Supplementary Fig. 5b), 13,951 particles were selected after particle auto-picking and manual checking, and 12,341 particles remained after reference-free 2D classification. We then deduced an initial model through ab initial reconstruction in cryoSPARC v2.15.0[52]. After further 3D classification of the dataset into four classes, we obtained three better classes revealing distinct configurations. After CTF refinement, Bayesian polishing, and refinement, these three classes

were subsequently refined to the CVA16−9B5 C1, C2, and C3 maps from 3122, 3967, and 3564 particles at the resolution of 2.90 Å, 3.35, and 3.80 Å, respectively. The overall resolutions for all of the cryo-EM maps in this study were determined based on the gold-standard criterion using a Fourier shell correlation of 0.143[53].

For the CVA16−8C4 dataset (Supplementary Fig. 7a), 20,196 particles were picked and 18,360 particles remained after reference-free 2D classification. We also obtained an initial model through ab initial reconstruction in cryoSPARC v2.15.0. We performed further heterogeneous refinement into five classes using the same initial model in cryoSPARC v2.15.0, which revealed three distinct major configurations. After CTF refinement, Bayesian polishing, and refinement, class1 and 3 were subsequently refined to the CVA16−8C4 C2 and C1 maps from 3690 and 2121 particles at the resolution of 2.91 and 3.05 Å, respectively. For Class 5, exhibiting open channels at the two-fold axis without bound antibody (in CVA16−8C4 C3 state), after further no-align 3D classification we obtained the class 3 showing better structural features, which was reconstructed to a map at 3.36-Å resolution after CTF refinement, Bayesian polishing, and refinement. All the obtained maps were post-processed through deepEMhancer[54].

**Atomic model building**
To build the atomic model for CVA16−9B5 and CVA16−8C4 maps, we used the corresponding cryo-EM structures of CVA16 from previous studies as initial model for the virial particle portion[16,17]. Specially, we used the full native CVA16 virion structure (PDB: 5C4W)[16] as the initial model for the C1 state of both CVA16−9B5 and CVA16−8C4, the natural empty CVA16 structure (PDB: 5C9A)[16] for C2 state, and the CVA16 empty particle structure (PDB: 6LHC)[17] for the C3 state. In the meanwhile, we built the homology models of 9B5 and 8C4 Fab through the SWISS-MODEL webserver[55]. We then combined the model and flexibly refined the model against the corresponding cryo-EM map utilizing Rossetta 2017[56], then Phenix 1.10.1[57]. Subsequently, to improve the fitting between model and map, we performed real-space refinement using COOT 0.8.3[58], then Phenix for the last round of flexible fitting of the entire complex. The final atomic models were validated by *phenix.molprobity* 1.10.1-2155[59]. The validation statistics of the atomic models were summarized in Supplementary Table 2.

Figures were generated using UCSF Chimera 1.10.2[60]. Fab-virion interaction analysis including hydrogen bond, salt bridge prediction, and buried surface area calculation were carried out through PISA server[61]. Roadmaps were generated by RIVEM 4.3 (Radial Interpretation of Viral Electron density Maps)[62]. Interaction surface analysis was conducted by using PDBePISA 1.48[63].

**Screening and sequencing of neutralization-resistant mutants**
To screen escape mutants, 100 TCID$_{50}$ of wild-type CVA16/SZ05 was incubated with various concentrations of MAb 8C4, 9B5, or 8C4 + 9B5 cocktail for 1 h at 37 °C before infection of RD cells in 96-well plates at 37 °C. After 4 days, the culture supernatants and cells were harvested and frozen and thawed twice, and 30 μl of the samples (passage 1 [P1] virus) were incubated with their selecting antibodies (10 μg/ml of MAb 8C4, 1 μg/ml of MAb 9B5, or the 8C4 + 9B5 cocktail [1 μg/ml of each MAb]) for 1 h at 37 °C prior to infection of fresh RD cells. After 3 days, 30 μl of the resultant P2 virus was subjected to another round of selection with their selecting MAbs (30 μg/ml of MAb 8C4, 10 μg/ml of MAb 9B5, or the 8C4 + 9B5 cocktail [10 μg/ml of each MAb]) for 1 h at 37 °C, yielding P3 virus. The resulting escape mutants were purified by plaque purification in RD cells overlaid with 0.4% low-melting point agarose (Promega) and 1% FBS in DMEM. For each MAb-selected mutant, 10 plaques were picked at 3 dpi and amplified in RD cells in 24-well plates. Total cellular RNA was extracted using TRIzol reagent (Invitrogen) and cDNA was the synthesized using M-MLV reverse transcriptase (Promega, USA). The capsid protein P1-coding region was amplified by PCR and then sequenced.

**Statistical analysis**
All statistical analyses were performed using GraphPad Prism version 8.

**Reporting summary**
Further information on research design is available in the Nature Portfolio Reporting Summary linked to this article.

## Data availability
Cryo-EM maps determined in the CVA16-9B5 dataset have been deposited in the Electron Microscopy Data Bank under accession codes: EMD-33941, EMD-34062, and EMD-34119, and the associated models have been deposited in the Protein Data Bank under accession codes: 7YMS, 7YRH, and 7YV7, respectively. Cryo-EM maps determined in the CVA16-8C4 dataset have been deposited in the Electron Microscopy Data Bank under accession codes: EMD-33670, EMD-34054, and EMD-34118, and the associated models have been deposited in the Protein Data Bank under accession codes: 7Y7M, 7YRF, and 7YV2, respectively. The sequences of 8C4-VH, 8C4-VL, 9B5-VH, and 9B5-VL have been deposited in GenBank under accession codes OP556479, OP556480, OP556481, and OP556482, respectively. All data analyzed during this study are included in the article. Source data are provided with this paper.

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

## Acknowledgements

We are grateful to the staffs of the NCPSS Electron Microscopy Facility, Database and Computing Facility, and Protein Expression and Purification Facility for instrument support and technical assistance. We thank Jiang Shao and the other staffs of the Institutional Center for Shared Technologies and Facilities of Institut Pasteur of Shanghai for instrument support and technical assistance. Z.H. was supported by grants from the National Natural Science Foundation of China (31872747), from the Chinese Academy of Sciences (XDB29040300), and from the Shanghai Municipal Science and Technology Major Project (ZD2021CY001). Y.C. was supported by the Strategic Priority Research Program of CAS (XDB37040103), the NSFC (32130056 and 31872714), National Key R&D Program of China (2017YFA0503503), Shanghai Academic Research Leader (20XD1404200), and Shanghai Pilot Program for Basic Research from CAS (JCYJ-SHFY-2022-008). C.Z. is supported by the Youth Innovation Promotion Association of CAS and Shanghai Rising-Star Program (21QA1410000).

## Author contributions

Z.H., Y.C., C.Z. and C.L. designed the experiments; C.Z., J.S., Y.W., X.Y., Q.L., X.L., W.Q. and Y.Y. performed the biochemical and animal experiments. C.L. collected the cryo-EM data and performed cryo-EM reconstructions and model building with the help of C.X. C.Z. and C.L. analyzed the data. Z.H., Y.C., C.Z. and C.L. wrote the manuscript.

## Competing interests

Z.H., J.S. and C.Z. are listed as inventors on pending patent applications for the MAbs. The other authors declare that they have no competing interests.
