## [Peer Review File · Nature Communications]

Review comments, first round -

Reviewer #1 (Remarks to the Author):

The manuscript by Zhang and colleagues describes two monoclonal antibodies (mAbs) against Coxsackievirus A16 (CVA16). CVA16 is an enterovirus and the major causative agent of hand, foot and mouth disease. Structurally, it follows the usual picornavirus structure with a protomer formed by VP0, VP1, and VP3, that assembles into an icosahedral capsid. The capsid proteins can adopt several conformations, including the mature capsid, the altered particle (A), a contracted empty capsid similar to the mature capsid (procapsid), and an expanded empty capsid. With the exception of the procapsid, in all the other capsid arrangements the VP0 is cleaved into VP2 and VP4, a small peptide located at the interior of the capsid. Importantly, a pocket factor located inside VP1, responsible of stabilizing the capsid, is also missing in the expanded empty capsid. The present study investigates two mAbs with different mode of action against CVA16 capsid. The authors perform a careful characterization of the mAbs: measure the mAbs neutralization and binding affinity; prove prophylactic efficacy and therapeutic efficacy for both mAbs; demonstrate that the mAbs have different modes of action; and perform a careful structural cryo-EM analysis of CVA16 virion decorated with Fabs. As expected, the antibodies recognize different epitopes: 9B5 is located around the five-fold axis, while 8C4 is located around the three-fold axis. For both mAbs. As expected from the capsid structures, the binding occurs for both the mature and contracted empty capsid. However, for the expanded empty capsid, only 9B5 binds to the capsid. The structures demonstrate the mode of action for the two mAbs: 9B5 masks the residues involved in the recognition of heparin sulphate – the attachment receptor for CVA16, and partially block the attachment of SCARB2 – the uncoating receptor; 8C4 blocks the attachment of SCARB2. Finally, the authors use serial passaging to identify escape mutants. They find mutations capable to block capsid binding for each mAb, but no resistance arises against the cocktail of both mAbs. The study is of very high quality; the text is clear and fluent; the figures are informative with adequate legends.

The findings are a solid and advance our structural knowledge of mAb neutralising mechanism of enteroviruses. Such mechanistic details are essential for successful development of therapies, such a viral inhibitors and vaccines.

The manuscript would be improved if the authors address the following points:

1. The authors have studied the therapeutic effect of the mAbs by administering them after CVA16 infection. Why did they choose the intervention moment at day one after infection? When do the first symptoms occur and how do they progress? While I don't recommend other experiments, it would be interesting if the authors would speculate on how late such a treatment could still function.
2. Is VP0 uncleaved in the compact empty capsid? Are other differences in that capsid region induced by mAb binding? If informative, I suggest a supplementary figure, similar to S4.
3. Other mAbs against CVA16 have been previously characterized. The authors attempt to analyse them together with their results and propose three classes of mAb. However, I consider that this analysis can be more consistent. For instance, the authors should present the footprints of different classes of mAbs with a RIVEM map. Also, they should mention their affinities, the recognized residues, the area of contact, etc. Maybe in a table.
4. The authors use a mouse adapted strain for their in vivo studies. Are all the residues involved in Ab recognition conserved in other circulating strains.
5. The study identifies a number of escape mutations. Are the identified residues conserved in different CVA16 strains?

Reviewer #2 (Remarks to the Author):

GENERAL COMMENTS:

CVA16 continues to cause disease outbreaks around the world. It's important to find effective, highly specific preventive and therapeutic interventions (such as neutralizing antibody). The manuscript reports the structural and functional analysis of two CVA16-specific neutralizing

antibodies targeting the 5- and 3-fold plateau. The central findings are that the investigators demonstrated the two antibodies adopt different mechanisms of action. The 9B5 exerted neutralization primarily through inhibiting CVA16 attachment to the cell surface via blockade of CVA16 binding to cellular receptor, including cell surface heparan sulfate and uncoating receptor SCARB2, whereas 8C4 cannot block virus attachment to cells and virus binding of heparan sulfate. However, 8C4 can functioned at the pre and post-attachment stage of CVA16 entry by interfering with the interaction between the virus and its uncoating receptor SCARB2. Then, they used cryo-EM and mapped that 9B5 and 8C4 exhibit the different binding patterns to the three CVA16 particle forms. Moreover, they also demonstrated that 9B5 and 8C4 antibodies were compatible in formulating an antibody cocktail which displayed increased neutralizing potency and ability to prevent virus escape. The neutralizing epitopes can guide the development of effective vaccines and immunotherapeutics against CVA16. Although this is an important viral pathogen and protective antibodies are presented, there are some question should be considered and revised.

SPECIAL COMMENTS:

In the neutralization mechanisms for 8C4 and 9B5, the description "9B5 exerted neutralization primarily through inhibiting CVA16 attachment to the cell surface via blockade of CVA16 binding to its attachment receptor, cell surface heparan sulfate, whereas 8C4 functioned mainly at the post-attachment stage of CVA16 entry by interfering with the interaction between the virus and its uncoating receptor SCARB2" in the abstract does not correspond to Fig. 3. As shown in fig.3A, 8C4 also exerts inhibitory effect at pre-attachment step (the relative viral RNA levels were determined to be 23.2%), why the authors described that 8C4 exerts inhibitory effect at an early post-attachment step. Previously, SCARB2 and PSGL1 have been identified to be the cellular receptors for CVA16. It's unknown whether 8C4 can prevent the interaction of CVA16 with other receptor during viral infection procedure. More evidence need to be provided to support this key point of view.

Previously, structural studies on EV71 or CVA16 show that naturally occurring EV71 or CVA16 particles mainly exist in two forms: mature virion and expanded empty particle. However, Ren et.al reported that formalin-treated CVA16 empty particle may exist in compact state. In this study, the authors demonstrate the existence of naturally occurring compact empty particle of CVA16. They should systematically compare and analyze the different strains used in the study and inactivation methods, then it may explain why CVA16 contain this particle form in this study.

The authors only showed that the IC50 against CVA16/SZ05 (Page5, Lines 116-118), how about the other two heterologous CVA16 strains (CVA16/GX08 and CVA16/MAV), they should add this data. In addition, the authors should describe more information about the three strains, such as genotypes.

In the binding properties of the anti-CVA16 MAbs, the experimental concentration setting of 9G1 antibody is unreasonable. When the concentration is greater than 12.5nM, the antibody is close to the saturated binding state, the author should add several groups of low antibody concentration groups, so as to more accurately calculate the KD value of the antibody.

In Fig. 1F-G, the results show that 9B5 can still bind after 9G1 binds first, but 9G1 cannot bind after 9B5 binds first. Thus, the authors described that 9B5 and 9G1 may share the same or highly overlapping binding sites on the CVA16 particle surface, which is not appropriate. The epitopes of the two antibodies may only partially overlap, or the angle of binding to the virus is different. So they should rewrite this part in a reasonable way and explain why the complex structure of 8C4 and virion was not resolved in the follow-up.

Field standard is to indicated mouse dosing of mAbs in terms of "mg/kg" rather than "µg/g." Even though these represent the same unit, would switch to the more standard "mg/kg" to avoid confusion. In addition, the unit of CPE or IC50 should be unified in fig.1A-B

One critical point of clarification is the increased neutralizing potency and ability to prevent virus escape by the 9B5 and 8C4 cocktail. However, the protective effects of MAbs 8C4 and 9B5 were assessed unreasonably. Regarding the 1:1 cocktails of mAbs for in vitro neutralization and in vivo

protection assays in Figures 6 and 3: When a dose is specified at 10µg/g for Figure 3, for example, is that 10 µg/g of EACH mAb, or a final concentration of 10 µg/g for the entire cocktail with each mAb individually at 5 µg/g? The former would not indicate synergy, but the latter would indicate synergy. The manuscript text makes me suspect the front. If so, I don't think it makes sense to design this way, and can't reflect whether the two antibody cocktails have better therapeutic activity in vivo.

For the mouse experiments in Figure 2, how many times did the authors repeat the in vivo experiment. Compared with the previous study (the mouse-adapted strain CVA16/MAV), the survival rate results of mice in the control group in this article are quite different. Please clarify. For mouse experiment replicates, I would include all data from all experiments aggregated into a single figure, or show all other experiments individually in supplemental files to show us if there is much variability between experiments and we are not simply being shown the best case scenario.

As shown in Fig. 3A, the addition of antibody 8C4 before virus adsorption (Pre) virus-bound cells were shifted to 37 °C resulted in significantly reduced viral RNA levels as compared to the virus-only group, the relative viral RNA levels were determined to be 23.2% for the 8C4 Pre group, suggesting that antibody 8C4 can exert inhibitory effect at pre-attachment step of the viral entry process. However, it's puzzled that 8C4 cannot inhibit CVA16 attachment to host cells in fig. 3B. It is recommended to explain the difference about the two experiments.

The authors suggest that the 8C4 Fab binds only to the native mature virion (C1) and compact empty particle (C2), rather than the expanded empty particle (C3) in Fig. 5A-C. Can they provide biochemical or immunological evidence for this? Furthermore, it is recommended to show details about the binding site of 8C4 among mature virion, compact empty particle and expanded empty particle, which may explain why 8C4 cannot binds to expanded empty particle (C3).

As shown in Fig. 6C-E, mutants resistant to antibody 8C4 or 9B5 rapidly emerged at the second or third passage, which may indicate the two antibodies are not suitable as potential treatment or prevent drugs. Thus, it is recommended to show the conservativeness of the binding sites of 8C4 and 9B5. In addition, the authors should explain that why 2.44 ng/mL of 9B5 cannot prevent rapid mutational escape, but the low concentration of 1.22 and 0.61 ng/mL of 9B5 can prevent the second and third mutational escape.

Reviewer #3 (Remarks to the Author):

This is an important work to functionally and mechanistically characterize two new CVA16 neutralizing mAbs, namely 9B5 and 8C4, isolated by the authors from antigen immunized mouse hybridomas. These two mAbs, which were non-competing in binding CVA16 particles, showed potent neutralization activity in cell culture assays and robust therapeutic efficacy in a lethal CVA16 challenge mouse model. The authors performed mechanistic studies to identify at what stage of the viral entry the two antibodies exert inhibitory function and to determine how the antibodies neutralize CVA16 (eg. by inhibiting virus attachment to cells and/or by blocking interactions between CVA16 and its host receptors including HS and SCARB2). The authors further performed cryo-EM analysis to show that 9B5 and 8C4 target non-overlapping epitopes located around the 5-fold and 3-fold axis of the viral capsid, respectively, with distinct binding preference to three naturally occurring CVA16 particle forms. Finally, the authors demonstrated that the 9B5/8C4 cocktail can prevent the occurrence of escape mutants which was observed for individual mAbs. Overall, the work in this paper is quite thorough and solid. The findings from this work represent an advance in the understanding of how antibodies can interact with and neutralize CVA16, a medically significant virus responsible for causing hand, foot and mouth disease, and may be useful in designing and producing prophylactic and therapeutic agents. The paper is generally well written and easy to follow.

Minor points:

- Fig.1E-G. It would be more convincing to add an irrelevant 1st antibody as the control in this

experiment.

- Supplementary Figure 2. In my opinion, the panels B and C are unclear and unnecessary and hence should be removed.

- Line 295. "Supplementary Table 5" should be changed to "Supplementary Table 4" according to the order of appearance.

- Line 340. "Supplementary Table 4" should be changed to "Supplementary Table 5".

Dear reviewers,

We would like to thank the reviewers for their insightful comments and suggestions. We have responded to each of the points raised by our reviewers and highlighted the changes in yellow in the revised text.

Response to reviewer #1's comments:

Reviewer #1 (Remarks to the Author):

The manuscript by Zhang and colleagues describes two monoclonal antibodies (mAbs) against Coxsackievirus A16 (CVA16). CVA16 is an enterovirus and the major causative agent of hand, foot and mouth disease. Structurally, it follows the usual picornavirus structure with a protomer formed by VP0, VP1, and VP3, that assembles into an icosahedral capsid. The capsid proteins can adopt several conformations, including the mature capsid, the altered particle (A), a contracted empty capsid similar to the mature capsid (procapsid), and an expanded empty capsid. With the exception of the procapsid, in all the other capsid arrangements the VP0 is cleaved into VP2 and VP4, a small peptide located at the interior of the capsid. Importantly, a pocket factor located inside VP1, responsible of stabilizing the capsid, is also missing in the expanded empty capsid.

The present study investigates two mAbs with different mode of action against CVA16 capsid. The authors perform a careful characterization of the mAbs: measure the mAbs neutralization and binding affinity; prove prophylactic efficacy and therapeutic efficacy for both mAbs; demonstrate that the mAbs have different modes of action; and perform a careful structural cryo-EM analysis of CVA16 virion decorated with Fabs. As expected, the antibodies recognize different epitopes: 9B5 is located around the five-fold axis, while 8C4 is located around the three-fold axis. For both mAbs. As expected from the capsid structures, the binding occurs for both the mature and contracted empty capsid. However, for the expanded empty capsid, only 9B5 binds to the capsid. The structures demonstrate the mode of action for the two mAbs: 9B5 masks the residues involved in the recognition of heparin sulphate – the attachment receptor for CVA16, and partially block the attachment of SCARB2 – the uncoating receptor; 8C4 blocks the attachment of SCARB2. Finally, the authors use serial passaging to identify escape mutants. They find mutations capable to block capsid binding for each mAb, but no resistance arises against the cocktail of both mAbs.

The study is of very high quality; the text is clear and fluent; the figures are informative with adequate legends.

The findings are a solid and advance our structural knowledge of mAb neutralising mechanism of enteroviruses. Such mechanistic details are essential for successful development of therapies, such a viral inhibitors and vaccines.

Response: We thank the reviewer for the positive comments. Below are our responses to the points raised by our reviewer.

The manuscript would be improved if the authors address the following points:

Q1-1. The authors have studied the therapeutic effect of the mAbs by administering

them after CVA16 infection. Why did they choose the intervention moment at day one after infection? When do the first symptoms occur and how do they progress? While I don't recommend other experiments, it would be interesting if the authors would speculate on how late such a treatment could still function.

A1-1: In this study we examined the therapeutic effect of the MAbs administered at one day post-infection (dpi), which is time point that we and other groups frequently used to assess the efficacy of anti-viral neutralizing MAbs ^{1, 2, 3}.

The mouse model of CVA16 infection has been described in detail in our previous paper ^{4, 5}. Typically, CVA16-infected mice began to exhibit clinical signs of disease (reduced mobility and limb weakness) at 3- or 4 days post infection (dpi) and subsequently developed limb paralysis, and some of these mice eventually died. Actually, in preliminary animal experiments we have assessed the therapeutic window of MAb 9B5 by administering different doses of 9B5 at 1, 3, or 5 dpi

[REDACTED]

Q1-2. Is VP0 uncleaved in the compact empty capsid? Are other differences in that capsid region induced by mAb binding? If informative, I suggest a supplementary figure, similar to S4.

A1-2: In our CVA16-8C4 C2 and CVA16-9B5 C2 structures, the VP4/VP2 junction region (amino acid K69^{VP4'} and S70^{VP2'} in VP0) are unresolved (**Fig. R2A**, for reviewers/editor's view only). Similarly, in the published structure (PDB: 5C9A) of CVA16 compact empty particle ⁶, the VP4/VP2 junction region was also unresolved. Thus, we are unable to conclude whether the VP0 in the CVA16 compact empty capsid is cleaved or not.

We have compared the capsid part of our different CVA16-8C4 and CVA16-9B5 states with the corresponding structures of three CVA16 particle forms without antibody binding, and no significant conformational difference was observed (**Fig. R2B-C**), suggesting that in our case the MAb binding does not induce conformational change in the viral capsid. As we have presented the structural comparison data (RMSD values) in Supplementary Table 3, we think it would be redundant to include Fig. R2 into the revised manuscript. Of course, if the reviewer/editor feels strongly about having it, we will be happy to do so.

Fig. R2 Structural comparison of the capsid proteins of the different CVA16-9B5 and CVA16-8C4 states with the known CVA16 particle structures. **(a)** The models of VP2 in the CVA16-8C4 C2 structure and the CVA16 compact empty particle structure (PDB: 5C9A). Unresolved residues are denoted with dashed lines. **(b-d)** Superposition of the protomers of the CVA16-9B5 C1-C3 structures (color) and the corresponding CVA16 particle structures (gray). **(e-g)** Superposition of the protomers of the CVA16-8C4 C1-C3 structures (color) and the corresponding CVA16 particle structures (gray).

Q1-3. Others mAbs against CVA16 have been previously characterized. The authors attempt to analyse them together with their results and propose three classes of mAb. However, I consider that this analysis can be more consistent. For instance, the authors should present the footprints of different classes of mAbs with a RIVEM map. Also, they should mention their affinities, the recognized residues, the area of contact, etc. Maybe in a table.

A1-3: Thanks for the constructive comment. As suggested, we have now presented the footprints of different classes of mAbs with a RIVEM map in the revised Supplementary Figure 9c (also shown below). We have also added related description in manuscript, to read “The binding footprints and the areas of contact of the three groups of anti-CVA16 antibodies are summarized in Supplementary Fig. 9c, d.” in lines 486-488.

Supplementary Figure 9. (c) Roadmap showing the footprints of the Fabs on the CVA16 virion surface. The 9B5, 18A7, NA9D7, 8C4, and 14B10 Fabs are indicated by purple, cyan, pink, yellow, and white contour lines, respectively. Surface area of CVA16 viral capsid covered by each Fab was also shown. **(d)** The surface area of contact between each Fab and CVA16 viral capsid.

The virus-binding affinity of the previously characterized MAbs 18A7, 14B10, and NA9D7 were not reported in the original paper ¹. The contacting residues of 18A7, 14B10, and NA9D7 were listed in Table S1 (Interactions between different Fabs and CVA16 mature virion) in that paper ¹. As this information can be easily obtained from the original paper, we feel it may not be necessary or appropriate to copy these data into a table in our manuscript. The areas of contact are ~1184, ~761, and ~1200 Å² for 18A7, 14B10, and NA9D7, respectively ¹, and as suggested we have incorporated the information into the new Supplementary Fig. 9d for comparison with our antibodies

9B5 and 8C4.

Q1-4. The authors use a mouse adapted strain for their in vivo studies. Are all the residues involved in Ab recognition conserved in other circulating strains.

A1-4: By sequence alignment, we found that the residues involved in 8C4 and 9B5 binding (Supplementary Table 4 and 6) were identical in the CVA16/SZ05 and CVA16/MAV (Genbank ID: KC695830) strains.

As suggested, we have analyzed the conservativeness of all the residues involved in 8C4 and 9B5 recognition in other CVA16 strains, and the results were shown in the revised Supplementary Table 4 and 6 (also shown below). We have also added related description in manuscript, to read “NCBI BLAST analysis revealed that the four contacting residues, G99, D104, R166 and K242, in VP1 are highly or fully conserved among the analyzed 1127 CVA16 VP1 sequences with identity of 99.6%, 98.0%, 100%, and 99.9%, respectively (Supplementary Table 4).” in lines 306-309, and “NCBI BLAST analysis showed that three contacting residues, K69, D74, and V159, in VP2 are extremely conserved (99.5% to 99.7%), while the other contacting residues in VP2 and VP3 are identical among all of the CVA16 strains analyzed (Supplementary Table 6).” in lines 363-366.

Supplementary Table 4. Interaction interface analysis of the CVA16–9B5 C1 structure (upper table) and conservation analysis of the contacting residues (lower table).

9B5-binding residues		Number of sequences	Conservation
VP1 (1127 sequences) ^a	G99	1122	99.6%
	D104	1105	98.0%
	R166	1127	100%
	K242	1126	99.9%

Supplementary Table 6. Interaction interface analysis of the CVA16–8C4 C1 structure (upper table) and C2 structure (middle table) and conservation analysis of the contacting residues (lower table).

8C4-binding residues		Number of sequences	Conservation
VP2 (399 sequences) ^b	K69 ^c	398	99.7%
	D74	397	99.5%
	W78	399	100%
	V159 ^c	397	99.5%

	G227	399	100%
	A228	399	100%
	S230	399	100%
	E231	399	100%
VP3	S77	191	100%
(191 sequences) ^b	T210	191	100%

^a Complete or near-complete CVA16 capsid protein sequences were obtained from NCBI database and used for conservation analysis.

^b The residues mutated in the neutralization escape mutants.

Q1-5. The study identifies a number of escape mutations. Are the identified residues conserved in different CVA16 strains?

A1-5: As suggested, we have analyzed the conservativeness of the residues mutated in the antibody-resistant mutants. We have added related description in manuscript, to read “Both VP2 V159 and K69 residues are extremely conserved (99.5% and 99.7% identity, respectively) among CVA16 strains (Supplementary Table 6), suggesting that they may be essential to CVA16 infectivity. Moreover, the 8C4-selected escape mutations V159F and K69Q are not present in all CVA16 strains.” in lines 433-437, and “NCBI BLAST analysis showed that VP1 T97, T100, and T103 are highly conserved across all CVA16 strains with conservation of 99.5%, 99.9%, and 98.2%, respectively. Moreover, the 9B5-selected escape mutations T97A+T100A or T100A+T103A are not present in all CVA16 strains.” in lines 443-446.

Response to reviewer #2' s comments:

Reviewer #2 (Remarks to the Author):

GENERAL COMMENTS:

CVA16 continues to cause disease outbreaks around the world. It's important to find effective, highly specific preventive and therapeutic interventions (such as neutralizing antibody). The manuscript reports the structural and functional analysis of two CVA16-specific neutralizing antibodies targeting the 5- and 3-fold plateau. The central findings are that the investigators demonstrated the two antibodies adopt different mechanisms of action. The 9B5 exerted neutralization primarily through inhibiting CVA16 attachment to the cell surface via blockade of CVA16 binding to cellular receptor, including cell surface heparan sulfate and uncoating receptor SCARB2, whereas 8C4 cannot block virus attachment to cells and virus binding of heparan sulfate. However, 8C4 can functioned at the pre and post-attachment stage of CVA16 entry by interfering with the interaction between the virus and its uncoating receptor SCARB2. Then, they used cryo-EM and mapped that 9B5 and 8C4 exhibit the different binding patterns to the three CVA16 particle forms. Moreover, they also demonstrated that 9B5 and 8C4 antibodies were compatible in formulating an antibody cocktail which displayed increased neutralizing potency and ability to prevent virus escape. The neutralizing epitopes can guide the development of effective vaccines and immunotherapeutics against CVA16. Although this is an important viral pathogen and protective antibodies are presented, there are some questions should be considered and revised.

Response: We thank the reviewer for the positive comments. Below are our responses to the reviewer's specific questions and comments.

SPECIAL COMMENTS:

Q2-1. In the neutralization mechanisms for 8C4 and 9B5, the description "9B5 exerted neutralization primarily through inhibiting CVA16 attachment to the cell surface via blockade of CVA16 binding to its attachment receptor, cell surface heparan sulfate, whereas 8C4 functioned mainly at the post-attachment stage of CVA16 entry by interfering with the interaction between the virus and its uncoating receptor SCARB2" in the abstract does not correspond to Fig. 3. As shown in fig.3A, 8C4 also exerts inhibitory effect at pre-attachment step (the relative viral RNA levels were determined to be 23.2%), why the authors described that 8C4 exerts inhibitory effect at an early post-attachment step. Previously, SCARB2 and PSGL1 have been identified to be the cellular receptors for CVA16. It's unknown whether 8C4 can prevent the interaction of CVA16 with other receptor during viral infection procedure. More evidence needs to be provided to support this key point of view.

A2-1: Fig.3A (also shown below for the reviewers' convenience) reports the results of the time-of-addition experiment in which the MAbs were added at different time points prior to or after virus infection and the readout is the relative viral RNA levels measured at 6 hours post-infection. According to Fig.3A, 8C4 was neutralizing to different degrees when added prior to virus attachment or at 0 or 0.5 h post-attachment (note

that, after virus attachment, the temperature of cell cultures was shifted to 37°C to allow viral entry). We should point out that the relative viral RNA levels of the 8C4-treated “post 0.5h” samples (49.6%) were significantly higher than those of the “pre” (23.2%) or the “post 0h” (29.8%) samples, indicating that 0.5 h post-attachment is a critical addition time point for 8C4 to mount neutralization. We reason that, at this time point (“post 0.5h”) a proportion of the cell-attached viruses may have been internalized and therefore cannot be accessed by 8C4 antibody, resulting in reduced neutralization effect. Nonetheless, to make it clear, we have now modified the statement (lines 215-220), to read “It was noted that the relative viral RNA levels in the 8C4 Post-0.5h group were significantly higher than those of the Pre or the Post-0h samples, indicating reduced neutralization effect by 8C4 administered at 0.5 h post-attachment. These results suggest that 0.5 h post-attachment is a critical addition time point for 8C4 to mount neutralization and therefore 8C4 may block an early step at the post-attachment stage of viral entry.”.

Fig.3 (a) Time of addition assay. 1000 TCID₅₀ of CVA16/SZ05 was exposed to the antibodies (1 µg of 8C4 or 9B5) before (pre-attachment, [Pre]) or at different time points after (post-attachment, [Post]) the virus attached to pre-chilled RD cells. Total RNA was isolated at 6 h after infection for real-time RT-PCR analysis of CVA16 RNA. **(b)** Attachment inhibition assay. 1.0 × 10⁶ TCID₅₀ of CVA16/SZ05 was incubated with various amounts of the MAbs (8C4, 9B5, or control [Ctr] antibody 2H2) for 1 h, and the mixtures were cooled and then allowed to attach to pre-chilled RD cells for 1 h at 4 °C. After rinse, total cellular RNA was extracted for real-time RT-PCR analysis of CVA16 RNA. In panels **a-b**, for each treatment, viral RNA levels relative to those for the only virus-infected samples are shown. Data are mean ± SEM of at least triplicate wells. Each symbol represents one well (24-well cell culture plate). Statistical significance between the virus-only and antibody-treated groups was calculated by t-test. ns, no significant difference ($p \geq 0.05$); *, $p < 0.05$; **, $p < 0.01$; ***, $p < 0.001$; ****, $p < 0.0001$.

Previous study has demonstrated that although several EV71 strains used PSGL-1 as the primary receptor for infection of Jurkat T cells, CVA16 infects lymphocytes (such as Jurkat T cells) independently of PSGL-1, because pretreatment with anti-PSGL-1 MAb KPL1 did not inhibit CVA16 infection in Jurkat T cells^{7,8}. Thus, PSGL-1 is not considered a functional receptor for CVA16, and is therefore not examined in the

present study.

Our group has previously shown that heparan sulfate (HS) glycosaminoglycans serve as an attachment receptor for CVA16⁹. In this study, we have demonstrated that 8C4 had no inhibitory effect on CVA16 binding to HS in the pull-down assay (Fig.3c) and on CVA16 attachment to the cell surface (Fig. 3b). Rather, we found that 8C4 could interfere with the interaction between CVA16 and SCARB2 (Fig.3d), which is further supported by structural observation (Fig.5m-n).

Q2-2. Previously, structural studies on EV71 or CVA16 show that naturally occurring EV71 or CVA16 particles mainly exist in two forms: mature virion and expanded empty particle. However, Ren et.al reported that formalin-treated CVA16 empty particle may exist in compact state. In this study, the authors demonstrate the existence of naturally occurring compact empty particle of CVA16. They should systematically compare and analyze the different strains used in the study and inactivation methods, then it may explain why CVA16 contain this particle form in this study.

A2-2: As suggested, we have analyzed the CVA16 strains used for structural analysis, including CVA16/SZ05, CVA16/Ningbo.CHN/028-2/2009, and CVA16/190, and the information was summarized in the below Table R1 (for the reviewers and editor's view only). The three strains differ from each other by less than 8 amino acids in capsid protein region. The CVA16 strains SZ05 and Ningbo.CHN/028-2/2009 were amplified in Vero cells and the strain 190 was amplified in RD cells. As for inactivation method, CVA16/SZ05 was incubated with MAbs 8C4 or 9B5 before structural analysis, Ningbo.CHN/028-2/2009 was inactivated with formaldehyde solution, whereas the strain 190 was live virus. From the comparison (Table R1), we are unable to extract a clear answer to why CVA16 contains the compact empty particle in our study.

[REDACTED]

Q2-3. The authors only showed that the IC50 against CVA16/SZ05 (Page5, Lines 116-118), how about the other two heterologous CVA16 strains (CVA16/GX08 and CVA16/MAV), they should add this data. In addition, the authors should describe more information about the three strains, such as genotypes.

A2-3: Thanks for the constructive suggestion. We have now determined the IC50 against CVA16/GX08 and CVA16/MAV and added the data into the revised Fig. 1A and Supplementary Fig. 2 (also shown below for the reviewers' convenient view). We have also added related description in manuscript, to read "IC50 of 8C4, 9B5, and 9G1 against CVA16/GX08 were determined to be 55, 1.2, and 7.9 ng/mL, respectively (Fig. 1a, Supplementary Fig. 2a)." in lines 129-130, and "IC50 of 8C4, 9B5, and 9G1 against CVA16/MAV were determined to be 52, 2.4, and 17 ng/mL, respectively (Fig. 1a, Supplementary Fig. 2b)." in lines 132-133.

MAb	Isotype	Neutralization concentration (ng/ml)				Neutralization IC50 (ng/mL)		
		CVA16/	CVA16/	CVA16/	EV71/	CVA16/	CVA16/	CVA16/
		SZ05	GX08	MAV	G082	SZ05	GX08	MAV
8C4	IgG2b	313	1250	1250	>100,000	74	55	52
9B5	IgG2b	1.2	4.9	4.9	>100,000	0.4	1.2	2.4
9G1	IgG2a	20	78	78	>100,000	4.9	7.9	17

Fig. 1. Neutralization, binding affinity and mutual competition of anti-CVA16 MAbs. **(a)** Isotypes and neutralization of anti-CVA16 MAbs (8C4, 9B5, and 9G1). Neutralization concentration was defined as the lowest antibody concentration that fully prevented cytopathic effect. Neutralization IC50 of each MAb against CVA16 was determined by the cell viability assay.

Supplementary Figure 2. Neutralization of the MAbs against CVA16 strains GX08 (a) and MAV (b) was measured by the cell viability assay. Related to Figure 1A. IgG-ctr, anti-SARS-CoV-2 MAb 2H2. Data are mean \pm SEM of five replicate wells in 96-well cell culture plates.

Q2-4. In the binding properties of the anti-CVA16 MAbs, the experimental concentration setting of 9G1 antibody is unreasonable. When the concentration is greater than 12.5nM, the antibody is close to the saturated binding state, the author should add several groups of low antibody concentration groups, so as to more accurately calculate the KD value of the antibody.

A2-4: As suggested, we have added low antibody concentration (0.78 and 0.20 nM) groups and repeated the experiment. Based on the new data (shown in the revised Fig. 1d and also below), the calculated KD value for 9G1 is 0.46 nM which is similar to the one (0.30 nM) determined previously. The related statement was modified accordingly (line 145), to read “As shown in Fig. 1D, all of the three MAbs exhibited high binding affinity to CVA16/SZ05 viral particles with equilibrium dissociation constants (KD) being 7.34, 0.35, and 0.46 nM for 8C4, 9B5, and 9G1, respectively.”.

Fig. 1. (d) Binding kinetics of the MAbs to immobilized CVA16/SZ05 viral particles were measured by biolayer interferometry (BLI). Association and dissociation steps are divided by dotted red line. MAb concentrations used were shown.

Q2-5. In Fig. 1F-G, the results show that 9B5 can still bind after 9G1 binds first, but 9G1 cannot bind after 9B5 binds first. Thus, the authors described that 9B5 and 9G1

may share the same or highly overlapping binding sites on the CVA16 particle surface, which is not appropriate. The epitopes of the two antibodies may only partially overlap, or the angle of binding to the virus is different. So they should rewrite this part in a reasonable way and explain why the complex structure of 9G1 and virion was not resolved in the follow-up.

A2-5: We agree with this reviewer's point. As suggested, we have modified the text in the revised manuscript accordingly (please see lines 162-163 and 166-167), to read "Together, these data show that the binding sites of 9B5 and 9G1 on the CVA16 particle surface are at least partially overlapping." and "MAb 9B5, which is a much more potent neutralizer than 9G1 in the group 1, and 8C4 in the group 2, were selected as the representatives of the two antibody groups for subsequent in-depth analyses".

Q2-6. Field standard is to indicated mouse dosing of mAbs in terms of "mg/kg" rather than "µg/g." Even though these represent the same unit, would switch to the more standard "mg/kg" to avoid confusion. In addition, the unit of CPE or IC50 should be unified in fig.1A-B

A2-6: As suggested, we have now replaced "µg/g" with "mg/kg in our revised manuscript.

Usually, we used "µg/mL" as the unit for IC50s, as in **Fig. 1B**. However, because the IC50 values of 9B5 were so small (0.4 to 2.4 ng/mL, or 0.0004 to 0.0024 µg/mL), we feel it is easier to use "ng/mL" rather than "µg/mL" as the unit in this case (**Fig. 1A**). Of course, we could change the unit to "µg/mL" if the editor/reviewer insisted.

Q2-7. One critical point of clarification is the increased neutralizing potency and ability to prevent virus escape by the 9B5 and 8C4 cocktail. However, the protective effects of MAbs 8C4 and 9B5 were assessed unreasonably. Regarding the 1:1 cocktails of mAbs for in vitro neutralization and in vivo protection assays in Figures 6 and 2: When a dose is specified at 10µg/g for Figure 3, for example, is that 10 µg/g of EACH mAb, or a final concentration of 10 µg/g for the entire cocktail with each mAb individually at 5 µg/g? The former would not indicate synergy, but the latter would indicate synergy. The manuscript text makes me suspect the front. If so, I don't think it makes sense to design this way, and can't reflect whether the two antibody cocktails have better therapeutic activity in vivo.

A2-7: We understand the reviewer's concern. Indeed, in the in vivo experiments (Figure 2) a single dose of the Mab cocktail contained 10 µg/g of EACH mAb. We designed such a dose to determine whether the two antibodies are compatible with each other when used in combination as a treatment in vivo. We did not intend to claim in the manuscript that the two-antibody combination has synergy or shows better therapeutic activity in vivo. As 9B5 alone has already provided 100% protection against CVA16 infection in mice (Figure 2), it would be difficult, if not impossible, to prove that the antibody cocktail is more potent than 9B5 in vivo.

Q2-8. For the mouse experiments in Figure 2, how many times did the authors repeat

the in vivo experiment. Compared with the previous study (the mouse-adapted strain CVA16/MAV), the survival rate results of mice in the control group in this article are quite different. Please clarify. For mouse experiment replicates, I would include all data from all experiments aggregated into a single figure, or show all other experiments individually in supplemental files to show us if there is much variability between experiments and we are not simply being shown the best case scenario.

A2-8: The mouse experiments (both prophylactic and therapeutic tests) were performed twice, and the results are quite similar. As suggested, we presented the datasets from the repeated experiments as the new Supplementary Fig. 3 (also shown below for the reviewers/editor's convenient view). In short, all of the mice administered anti-CVA16 MAbs, except one 8C4-treated mouse in the therapeutic test, were completely protected. In contrast, the mice in the PBS or the control IgG groups displayed significant mortality (please see the below Supplementary Fig. 3c). We have modified the text in the revised manuscript accordingly (lines 196-202), to read "To confirm the in vivo efficacy of the anti-CVA16 MAbs, both prophylactic and therapeutic tests were repeated and similar results were obtained (Supplementary Fig. 3). All of the mice administered anti-CVA16 MAbs, except one 8C4-treated mouse in the therapeutic test, were completely protected. In contrast, the mice in the PBS or the control IgG groups displayed significant mortality ranging from 46% to 67% (Supplementary Fig. 3c). Collectively, the above data demonstrate that 8C4 and 9B5 possess robust prophylactic and therapeutic efficacies in vivo."

Supplementary Figure 3. In vivo prophylactic efficacy (a) and therapeutic efficacy (b) of MAbs 8C4, 9B5 and the 8C4+9B5 cocktail against CVA16 infection in mice. Results of two independent experiments are shown in Figure 2 and Supplementary Figure 3, respectively. Clinical scores were graded as follows: 0, healthy; 1, reduced mobility; 2, limb weakness; 3, limb paralysis; 4, death. Survival rates of MAb-treated mice were compared with the mice in the PBS-treated group. Statistical significance was determined by Log-rank (Mantel-Cox) test. ns, no significant difference ($p \geq 0.05$); **, $p < 0.01$; ***, $p < 0.001$. All error bars represent SEM. (c) The survival rates of mice in PBS and IgG-ctr groups.

Q2-9. As shown in Fig. 3A, the addition of antibody 8C4 before virus adsorption (Pre) virus-bound cells were shifted to 37 °C resulted in significantly reduced viral RNA levels as compared to the virus-only group, the relative viral RNA levels were determined to be 23.2% for the 8C4 Pre group, suggesting that antibody 8C4 can exert inhibitory effect at pre-attachment step of the viral entry process. However, it's puzzled that 8C4 cannot inhibit CVA16 attachment to host cells in fig. 3B. It is recommended to explain the difference about the two experiments.

A2-9: In time-of-addition experiments, the latest addition time point when a given

drug's effectiveness starts to drop or diminish is considered the critical time point for this drug to exert its function. That is to say, the drug likely takes action immediately prior to this time point. In the case of 8C4 (Fig. 3A), the relative viral RNA levels of the "post-0.5h" sample were significantly higher (49.6%) than those of the "pre" (23.2%) and the "post-0h" (29.8%) samples; in another word, the neutralization efficiency of 8C4 dropped significantly when administered at the post-0.5h time point, indicating that 8C4 exerts inhibitory function prior to this time point. In addition, the relative viral RNA levels of the 8C4-treated, "pre" samples are not statistically different from those of the "post-0h" samples, suggesting that 8C4 does not prevent virus attachment. This speculation was verified by the attachment-inhibition experiment (Fig. 3B). To better explain the difference between the two experiments, we have pasted below the Figure 3 and also provided the flowcharts of the experiments in the below Fig. R3 (for the reviewers/editor's view only). Based on our data (Fig. 3A and 3B), we conclude that 8C4 may neutralize CVA16 by inhibiting an early (e.g. <0.5 h) post-attachment step of the viral entry process. As to the question "However, it's puzzled that 8C4 cannot inhibit CVA16 attachment to host cells in fig. 3B.", we reason that, 8C4, when added prior to virus attachment, can still bind the virus, and the resulting 8C4/virus complex can still attach to the cells and be internalized but it is prohibited from uncoating due to blockade of the CVA16/SCARB2 interaction by 8C4 (Fig. 3d and Fig. 5m-n). We hope the reviewer will be satisfied with our explanation.

Fig R3. The flowcharts of the time of addition assay and attachment inhibition assay.

Fig.3 (a) Time of addition assay. 1000 TCID_{50} of CVA16/SZ05 was exposed to the antibodies ($1 \mu\text{g}$ of 8C4 or 9B5) before (pre-attachment, [Pre]) or at different time points after (post-attachment, [Post]) the virus attached to pre-chilled RD cells. Total RNA was isolated at 6 h after infection for real-time RT-PCR analysis of CVA16 RNA. **(b)** Attachment inhibition assay. $1.0 \times 10^6 \text{ TCID}_{50}$ of CVA16/SZ05 was incubated with various amounts of the MAbs (8C4, 9B5, or control [Ctr] antibody 2H2) for 1 h, and the mixtures were cooled and then allowed to attach to pre-chilled RD cells for 1 h at 4°C . After rinse, total cellular RNA was extracted for real-time RT-PCR analysis of CVA16 RNA. In panels **a-b**, for each treatment, viral RNA levels relative to those for the only virus-infected samples are shown. Data are mean \pm SEM of at least triplicate wells. Each symbol represents one well (24-well cell culture plate). Statistical significance between the virus-only and antibody-treated groups was calculated by t-test. ns, no significant difference ($p \geq 0.05$); *, $p < 0.05$; **, $p < 0.01$; ***, $p < 0.001$; ****, $p < 0.0001$.

Q2-10. The authors suggest that the 8C4 Fab binds only to the native mature virion (C1) and compact empty particle (C2), rather than the expanded empty particle (C3) in Fig. 5A-C. Can they provide biochemical or immunological evidence for this? Furthermore, it is recommended to show details about the binding site of 8C4 among mature virion, compact empty particle and expanded empty particle, which may explain why 8C4 cannot binds to expanded empty particle (C3).

A2-10: We appreciate the comments. Unfortunately, the three forms of CVA16 particles cannot be efficiently separated by sucrose gradient ultracentrifugation in our study, so we cannot use biochemical or immunological approaches to determine the binding preference of 8C4 to the three particle forms.

The details about the binding site of 8C4 on the mature virion have been provided in Fig. 5j-n and in Supplementary Table 4 (the revised Supplementary Table 6) in our previous submission. As suggested, we have now analyzed the 8C4 binding site on compact empty particle and expanded empty particle (please see the new Supplementary Fig. 8, also shown below for convenient view). In the CVA16-8C4 C2 structure, the conformation of 8C4 binding site on the capsid is maintained to well accommodate the 8C4 Fab, despite both of the antibody and capsid protein display a very slight upward movement relative to the counterparts in the CVA16-8C4 C1 model

(new Supplementary Fig. 8a). Further analysis reveals that, compared to CVA16-8C4 C1, the CVA16-8C4 C2 complex preserves most of the interactions between 8C4 and viral capsid (Supplementary Table 6). In contrast, compared with CVA16-8C4 C1, CVA16-8C4 C3 displays an obvious outward tilting movement of up to 5.2 Å and 4.7 Å towards the Fab for the protomer 1 and protomer 2, respectively (new Supplementary Fig. 8b). Such a large movement towards the 8C4 Fab may create a clash between the capsid and the 8C4 heavy chain (new Supplementary Fig. 8b). This potentially explained why 8C4 failed to bind CVA16 expanded empty particle.

We have modified the text in the revised manuscript accordingly (lines 370-382), to read “In the CVA16-8C4 C2 structure, the conformation of 8C4 binding site on the capsid is well maintained to accommodate the 8C4 Fab, despite both of the antibody and capsid protein display a very slight upward movement relative to the counterparts in the CVA16-8C4 C1 model (Supplementary Fig. 8a). Further analysis reveals that, compared to CVA16-8C4 C1, the C2 complex preserves most of the interactions between 8C4 and viral capsid (Supplementary Table 6). In contrast, compared with CVA16-8C4 C1, CVA16-8C4 C3 displays a quite large upward movement for the expanded capsid and in particular an obvious outward tilting movement of up to 5.2 Å and 4.7 Å towards the Fab for the protomer 1 and protomer 2, respectively (Supplementary Fig. 8b). Such a large movement towards the 8C4 Fab may create clashes between the CVA16 capsid and the 8C4 heavy chain, thus prevent 8C4 binding (Supplementary Fig. 8b). This potentially explained why 8C4 failed to bind CVA16 expanded empty particle.”

Supplementary Figure 8. Structural basis of antibody 8C4 binding to CVA16 capsid in the C2 structure but not in C3. (a) Superposition of the CVA16-8C4 C1 (gray) and C2 (color) structures. The red circles indicate slight movements of the capsid in the C2 structure relative to the C1 structure. (b) Docking of the C3 structure (magenta) into the C1 structure (gray). The red arrow represents the movement direction of the capsid in the C3 structure relative to the C1 structure.

Supplementary Table 6. Interaction interface analysis of the CVA16–8C4 C1 structure (upper table) and C2 structure (middle table) and conservation analysis of the contacting residues (lower table).

C1	CVA16	Distance	8C4	Interaction
----	-------	----------	-----	-------------

structure	Residue	Location	(Å)	Residue	Location	
Protomer 1	S77 [O]	VP3 BC loop	3.17	R99 [NH2]	HCDR3	H-bond
	S77 [O]	VP3 BC loop	3.32	R99 [NH1]	HCDR3	H-bond
	T210 [OG1]	VP3 HI loop	3.07	N30 [OD1]	LCDR1	H-bond
Protomer 2	S230 [OG]	VP2 HI loop	3.28	W92 [O]	LCDR3	H-bond
	G227 [O]	VP2 HI loop	3.07	W92 [NE1]	LCDR3	H-bond
	A228 [O]	VP2 HI loop	3.43	W92 [NE1]	LCDR3	H-bond
	K69 [NZ]	VP2 βB	2.73	D55 [OD2]	HCDR2	H-bond, salt bridge
	W78 [NE1] ^a	VP2 βC	3.55	D57 [OD2]	HCDR2	H-bond
	V159 [N]	VP2 EF loop	3.49	D55 [O]	HCDR2	H-bond
	D74 [OD1]	VP2 BC loop	3.73	Y60 [N]	HFR3	H-bond
	D74 [OD1]	VP2 BC loop	3.60	N59 [ND2]	HFR3	H-bond
	D74 [OD2]	VP2 BC loop	3.72	N59 [ND2]	HFR3	H-bond
	E231 [OE1] ^a	VP2 HI loop	3.22	R99 [NH2]	HCDR3	H-bond, salt bridge

C2 structure	CVA16		Distance (Å)	8C4		Interaction
	Residue	Location		Residue	Location	
Protomer 1	S77 [O]	VP3 BC loop	3.61	R99 [NH2]	HCDR3	H-bond
	S77 [O]	VP3 BC loop	3.51	R99 [NH1]	HCDR3	H-bond
	T210 [OG1]	VP3 HI loop	2.83	N30 [ND2]	LCDR1	H-bond
Protomer 2	S230 [OG]	VP2 HI loop	2.82	W92 [O]	LCDR3	H-bond
	S230 [N]	VP2 HI loop	3.55	N93 [OD1]	LCDR3	H-bond
	G227 [O]	VP2 HI loop	3.28	W92 [NE1]	LCDR3	H-bond
	A228 [O]	VP2 HI loop	3.49	W92 [NE1]	LCDR3	H-bond
	K69 [NZ]	VP2 βB	2.70	D55 [OD2]	HCDR2	H-bond, salt bridge
	V159 [N]	VP2 EF loop	3.58	D55 [O]	HCDR2	H-bond
	D74 [OD1]	VP2 BC loop	2.64	Y60 [N]	HFR3	H-bond
	D74 [OD1]	VP2 BC loop	2.87	N59 [ND2]	HFR3	H-bond

^a The interaction exists in the C1 structure, but not in the C2 structure.

Q2-11. As shown in Fig. 6C-E, mutants resistant to antibody 8C4 or 9B5 rapidly emerged at the second or third passage, which may indicate the two antibodies are not suitable as potential treatment or prevent drugs. Thus, it is recommended to show

the conservativeness of the binding sites of 8C4 and 9B5. In addition, the authors should explain that why 2.44 ng/mL of 9B5 cannot prevent rapid mutational escape, but the low concentration of 1.22 and 0.61 ng/mL of 9B5 can prevent the second and third mutational escape.

A2-11: As suggested, we have analyzed the conservativeness of the binding sites of 8C4 and 9B5, and the results were shown in the revised Supplementary Table 4 and 6 (also shown below for the reviewers/editor’s convenient view). We have also added related description in manuscript, to read “NCBI BLAST analysis revealed that the four contacting residues, G99, D104, R166 and K242, in VP1 are highly or fully conserved among the analyzed 1127 CVA16 VP1 sequences with identity of 99.6%, 98.0%, 100%, and 99.9%, respectively (Supplementary Table 4).” in lines 306-309, and “NCBI BLAST analysis showed that three contacting residues, K69, D74, and V159, in VP2 are extremely conserved (99.5% to 99.7%), while the other contacting residues in VP2 and VP3 are identical among all of the CVA16 strains analyzed (Supplementary Table 6).” in lines 363-366.

Supplementary Table 4. Interaction interface analysis of the CVA16–9B5 C1 structure (upper table) and conservation analysis of the contacting residues (lower table).

9B5-binding residues		Number of sequences	Conservation
VP1 (1127 sequences) ^a	G99	1122	99.6%
	D104	1105	98.0%
	R166	1127	100%
	K242	1126	99.9%

Supplementary Table 6. Interaction interface analysis of the CVA16–8C4 C1 structure (upper table) and conservation analysis of the contacting residues (lower table).

8C4-binding residues		Number of sequences	Conservation
VP2 (399 sequences) ^a	K69 ^b	398	99.7%
	D74	397	99.5%
	W78	399	100%
	V159 ^b	397	99.5%
	G227	399	100%
	A228	399	100%

	S230	399	100%
	E231	399	100%
VP3	S77	191	100%
(191 sequences) ^a	T210	191	100%

^a Complete or near-complete CVA16 capsid protein sequences were obtained from NCBI database and used for conservation analysis.

^b The residues mutated in the neutralization escape mutants.

It is well known that, for a given antibody, an appropriate antibody concentration is very important for successful selection of escape mutants¹⁰: if the antibody concentration/pressure was too high, all of the virus would be neutralized; if it was too low, the wildtype virus would remain infectious and dominant and escape mutation would not occur under such pressure. In the case of 9B5, 2.44 ng/mL appeared to be the appropriate antibody concentration for selection of 9B5-resistant mutants, whereas higher concentrations (>2.44 ng/mL) of 9B5 resulted in complete neutralization of CVA16 and lower concentrations (0.61 or 1.22 ng/mL) did not sufficiently pressure the virus to undergo directional mutation. We hope the reviewer will be satisfied with our explanation.

Response to reviewer #3' s comments:

Reviewer #3 (Remarks to the Author):

This is an important work to functionally and mechanistically characterize two new CVA16 neutralizing Mabs, namely 9B5 and 8C4, isolated by the authors from antigen immunized mouse hybridomas. These two mAbs, which were non-competing in binding CVA16 particles, showed potent neutralization activity in cell culture assays and robust therapeutic efficacy in a lethal CVA16 challenge mouse model. The authors performed mechanistic studies to identify at what stage of the viral entry the two antibodies exert inhibitory function and to determine how the antibodies neutralize CVA16 (eg. by inhibiting virus attachment to cells and/or by blocking interactions between CVA16 and its host receptors including HS and SCARB2). The authors further performed cryo-EM analysis to show that 9B5 and 8C4 target non-overlapping epitopes located around the 5-fold and 3-fold axis of the viral capsid, respectively, with distinct binding preference to three naturally occurring CVA16 particle forms. Finally, the authors demonstrated that the 9B5/8C4 cocktail can prevent the occurrence of escape mutants which was observed for individual mAbs. Overall, the work in this paper is quite thorough and solid. The findings from this work represent an advance in the understanding of how antibodies can interact with and neutralize CVA16, a medically significant virus responsible for causing hand, foot and mouth disease, and may be useful in designing and producing prophylactic and therapeutic agents. The paper is generally well written and easy to follow.

Response: Thanks for the encouraging comments from our reviewer.

Minor points:

Q3-1. Fig.1E-G. It would be more convincing to add an irrelevant 1st antibody as the control in this experiment.

A3-1: We have followed the suggestion to include an irrelevant 1st antibody as the control in the antibody competition experiment. As expected, compared with buffer alone, pre-incubation with the control IgG antibody 2H2 did not affect the binding of 8C4, 9B5, and 9G1 to CVA16 virion, while the anti-CVA16 MAbs (8C4, 9B5, and 9G1) showed binding competition profiles similar to the ones presented in our previous submission. The new dataset was now provided as the revised Fig. 1e-g and also shown below for the reviewers' convenient view. We have also added related description in manuscript, to read "Compared with buffer alone, pre-incubation with the control IgG antibody 2H2 did not affect the binding of 8C4, 9B5, or 9G1 to CVA16 virion." in lines 1044-1045.

Fig. 1. (e-g) BLI-based antibody competition assay. Immobilized CVA16/SZ05 viral particles were first incubated with buffer (reference) or the indicated MAb (first antibody) and then incubated with the second MAb 8C4 (e), 9B5 (f), or 9G1 (g) in the presence of the first MAb. An irrelevant MAb 2H2 served as the control antibody (gray curve) in the assay. The graphs show binding signals of the second MAb 8C4 (e), 9B5 (f), and 9G1 (g).

Q3-2. Supplementary Figure 2. In my opinion, the panels B and C are unclear and unnecessary and hence should be removed.

A3-2: As suggested, we have deleted the panels B and C in the previous Supplementary Figure 2 (now renamed Supplementary Figure 4).

Q3-3. Line 295. “Supplementary Table 5” should be changed to “Supplementary Table 4” according to the order of appearance.

A3-3: We have re-arranged the order of supplementary tables accordingly. The previous “Supplementary Table 4” is now “Supplementary Table 6”.

Q3-4.- Line 340. “Supplementary Table 4” should be changed to “Supplementary Table 5”.

A3-4: Done as suggested.

Reference

1. He M, *et al.* Identification of Antibodies with Non-overlapping Neutralization Sites that Target Coxsackievirus A16. *Cell Host Microbe* **27**, 249–261 e245 (2020).
2. Zhang C, *et al.* Functional and structural characterization of a two-MAb cocktail for delayed treatment of enterovirus D68 infections. *Nat Commun* **12**, 2904 (2021).
3. Zhang C, *et al.* Development and structural basis of a two-MAb cocktail for treating SARS-CoV-2 infections. *Nat Commun* **12**, 264 (2021).
4. Cai Y, *et al.* Active immunization with a Coxsackievirus A16 experimental inactivated vaccine induces neutralizing antibodies and protects mice against lethal infection. *Vaccine* **31**, 2215–2221 (2013).
5. Liu Q, *et al.* A murine model of coxsackievirus A16 infection for anti-viral

- evaluation. *Antiviral Res* **105**, 26-31 (2014).
6. Ren J, *et al.* Structures of Coxsackievirus A16 Capsids with Native Antigenicity: Implications for Particle Expansion, Receptor Binding, and Immunogenicity. *J Virol* **89**, 10500-10511 (2015).
 7. Nishimura Y, Shimojima M, Tano Y, Miyamura T, Wakita T, Shimizu H. Human P-selectin glycoprotein ligand-1 is a functional receptor for enterovirus 71. *Nat Med* **15**, 794-797 (2009).
 8. Patel KP, Bergelson JM. Receptors identified for hand, foot and mouth virus. *Nat Med* **15**, 728-729 (2009).
 9. Zhang X, *et al.* Coxsackievirus A16 utilizes cell surface heparan sulfate glycosaminoglycans as its attachment receptor. *Emerg Microbes Infect* **6**, e65 (2017).
 10. Baum A, *et al.* Antibody cocktail to SARS-CoV-2 spike protein prevents rapid mutational escape seen with individual antibodies. *Science* **369**, 1014-1018 (2020).

Review comments, second round -

Reviewer #1 (Remarks to the Author):

I consider that the authors have convincingly responded to all the points raised by the reviewers.

Reviewer #2 (Remarks to the Author):

The authors have successfully addressed many concerns I raised and the manuscript is now obviously improved. There still remains three concerns which should be addressed:
In the response manuscript questions 2, whether the authors analysis the CVA16-SZ05 contain the naturally occurring compact empty particle without incubation with MAb.
The authors did not describe the information about the three strains, such as genotypes in questions 3.
Questions 7, according to the authors response, they did not intend to claim in the manuscript that the two-antibody combination has synergy. Thus, they should revised the describe of "we demonstrated that 9B5 and 8C4 antibodies were compatible in formulating an antibody cocktail which displayed increased neutralizing potency" in the abstract, as they did not provide data to support this conclusion.

Reviewer #3 (Remarks to the Author):

The authors have satisfactorily addressed all my concerns and revised the manuscript accordingly. Therefore, I recommend publication of this manuscript.

Response to reviewer #1's comments:

Reviewer #1 (Remarks to the Author):

I consider that the authors have convincingly responded to all the points raised by the reviewers.

Response: Thanks.

Reviewer #2 (Remarks to the Author):

The authors have successfully addressed many concerns I raised and the manuscript is now obviously improved. There still remain three concerns which should be addressed:

1. In the response manuscript questions 2, whether the authors analysis the CVA16-SZ05 contain the naturally occurring compact empty particle without incubation with MAb.

Response: This question is not very clear. The reviewer is probably asking whether we have performed cryo-EM to determine whether infectious CVA16 preparation (without incubation with MAbs) contains the naturally occurring compact empty particle form. The answer is that we did not perform cryo-EM on live CVA16 preparation without MAb neutralization, because the cryo-EM facility in the National Center for Protein Science Shanghai (NCPSS) is not a biosafety level 2 (BSL2) facility and therefore does not allow direct examination of live virus samples that have not been properly inactivated or neutralized.

2. The authors did not describe the information about the three strains, such as genotypes in questions 3.

Response: The information about the three CVA16 strains, including genotype and GenBank ID, has been added into the Supplementary Table 1 (also shown below for the reviewers' convenience).

Supplementary Table 1. A summary of all the CVA16 strains used in this study.

CVA16 strains	Genotype	Genbank ID
CVA16/SZ05 ^a	B1b	EU262658
CVA16/GX08	B1b	KC342228
CVA16/MAV	B1a	KC695830

^a used as the antigen for mouse immunization.

3. Questions 7, according to the authors response, they did not intend to claim in the manuscript that the two-antibody combination has synergy. Thus, they should revise

the describe of “we demonstrated that 9B5 and 8C4 antibodies were compatible in formulating an antibody cocktail which displayed increased neutralizing potency” in the abstract, as they did not provide data to support this conclusion.

Response: The point is taken. We have now revised the statement, to read “9B5 and 8C4 are compatible in formulating an antibody cocktail which displays the ability to prevent virus escape seen with individual MAbs.” in the abstract.

Reviewer #3 (Remarks to the Author):

The authors have satisfactorily addressed all my concerns and revised the manuscript accordingly. Therefore, I recommend publication of this manuscript.

Response: Thanks.